# Detection of PatIent-Level distances from single cell genomics and pathomics data with Optimal Transport (PILOT)

Mehdi Joodaki[1,10], Mina Shaigan [ID][1,10], Victor Parra[1], Roman D Bülow [ID][2], Christoph Kuppe [ID][3], David L Hölscher [ID][2], Mingbo Cheng [ID][1], James S Nagai [ID][1], Michaël Goedertier[1,2], Nassim Bouteldja[2], Vladimir Tesar [ID][4], Jonathan Barratt[5,6], Ian SD Roberts [ID][7], Rosanna Coppo[8], Rafael Kramann [ID][3,9], Peter Boor [ID][2✉] & Ivan G Costa [ID][1✉]

## Abstract

**Although clinical applications represent the next challenge in single-cell genomics and digital pathology, we still lack computational methods to analyze single-cell or pathomics data to find sample-level trajectories or clusters associated with diseases. This remains challenging as single-cell/pathomics data are multi-scale, i.e., a sample is represented by clusters of cells/structures, and samples cannot be easily compared with each other. Here we propose PatIent Level analysis with Optimal Transport (PILOT). PILOT uses optimal transport to compute the Wasserstein distance between two individual single-cell samples. This allows us to perform unsupervised analysis at the sample level and uncover trajectories or cellular clusters associated with disease progression. We evaluate PILOT and competing approaches in single-cell genomics or pathomics studies involving various human diseases with up to 600 samples/patients and millions of cells or tissue structures. Our results demonstrate that PILOT detects disease-associated samples from large and complex single-cell or pathomics data. Moreover, PILOT provides a statistical approach to find changes in cell populations, gene expression, and tissue structures related to the trajectories or clusters supporting interpretation of predictions.**

**Keywords** Clustering; Disease Progression; Multi-scale Analysis; Optimal Transport; Wasserstein Distance
**Subject Categories** Chromatin, Transcription & Genomics; Computational Biology

## Introduction

Single-cell genomics and digital pathology methods are revolutionary technologies, which in principle, allow researchers to computationally dissect molecular, cellular, and structural changes in human tissues (Cao et al, 2020; Consortium et al, 2022). However, the clinical application of single-cell sequencing, i.e., finding cells and their markers for patient stratification and personalized treatments (Sklavenitis-Pistofidis et al, 2022), is still in its infancy. Recent clinical genomics efforts include the use of single-cell transcriptomics to dissect the progression of acute human myocardial infarction (Kuppe et al, 2020), to characterize COVID-19 patients' disease severity (Stephenson et al, 2021), the development of pancreatic ductal adenocarcinomas (Peng et al, 2019) or to study Alzheimer disease (Cain et al, 2023), just to cite a few. Computational analysis of these datasets mainly leverages standard methods for standard single-cell sequencing analysis, i.e., finding genes with differential expression in control vs. disease for each cell cluster. These approaches require a-priori patient classifications, i.e., control vs. disease. Therefore, they cannot be used to find novel subgroups of patients. Alternatively, trajectory analysis can be performed to uncover disease progression allowing the characterization of early disease events (Chen et al, 2020). Particularly challenging is the multi-scale nature of single-cell experiments, i.e., each sample is represented by thousands of single cells, which are clustered in distinct cell types or cell states. Currently, there are few analytical methods to compare multiple single-cell experiments from the same tissue from multiple distinct individuals.

Pathomics, i.e., the use of machine learning methods to extract morphological structures in histology slides (Bülow et al, 2022; Hölscher et al, 2023), is a technique orthogonal to single-cell data, which also generated multi-scale data. Pathomics data of a slide is represented by thousands of anatomical structures, which are described by morphometric features, e.g., their individual shape and size. These can be clustered to find structures at distinct morphological states, i.e., distinct

---

[1]Institute for Computational Genomics, Joint Research Center for Computational Biomedicine, RWTH Aachen University Medical School, Aachen, Germany. [2]Institute of Pathology, RWTH Aachen University Medical School, Aachen, Germany. [3]Institute of Experimental Medicine and Systems Biology, RWTH Aachen University, Aachen, Germany. [4]Department of Nephrology, 1st Faculty of Medicine and General University Hospital, Charles University, Prague, Czech Republic. [5]John Walls Renal Unit, University Hospital of Leicester National Health Service Trust, Leicester, UK. [6]Department of Cardiovascular Sciences, University of Leicester, Leicester, UK. [7]Department of Cellular Pathology, Oxford University Hospitals National Health Services Foundation Trust, Oxford, UK. [8]Fondazione Ricerca Molinette, Regina Margherita Children's University Hospital, Torino, Italy. [9]Department of Internal Medicine, Nephrology and Transplantation, Erasmus Medical Center, Rotterdam, Netherlands. [10]These authors contributed equally as first authors: Mehdi Joodaki, Mina Shaigan. ✉E-mail: pboor@ukaachen.de; ivan.costa@rwth-aachen.de

degrees of dysmorphism. There are limited computational methods to compare two or more histological slides based on morphometric properties of their structures.

Until now, only a few methods allow the analysis of single-cell genomics datasets at a sample level. PhEMD (Chen et al, 2020) is based on earth moving distance (EMD) to measure the distance between specimens (single-cell samples), where the distance between specimens was based on clustering representations from a diffusion-based space. This method was successfully used to explore the response of cell lines to drug effects. Nevertheless, it is based on the diffusion map and pseudotime estimates on the cell level. This explicitly assumes the presence of a cellular continuum (cell differentiation/activation) between all cells in the scRNA-seq experiments. Therefore, it is not suitable for the analysis of single-cell experiments measured in whole organ samples with heterogeneous cell populations. Moreover, PhEMD lacks methods for interpretation of the results, i.e., detection of molecular and cellular features explaining predictions. Recently, Flores and colleagues proposed a multi-omics factor analysis that analyses single-cell data at pseudo-bulk level (Flores et al, 2023). This work focus on finding factors and molecular features explaining previously known sample phenotypes. Ravindra and colleagues (Ravindra et al, 2020) propose the use of graph attention networks for the classification of scRNA-seq samples and apply this to predict the disease states of multiple Sclerosis (Lublin et al, 1996). More recently, SCANCell (Zhang et al, 2022), which uses PhEMD to generate non-linear embeddings followed by the use of association networks across cell clusters, was proposed and applied for the analysis of systemic lupus erythematodes. Multi-scale PHATE (Kuchroo et al, 2022) is based on non-linear embeddings and multi-resolution representation to cluster cells at distinct resolutions. Moreover, it uses the proportion of cells per sample (across distinct resolutions) and PHATE non-linear embeddings (Moon et al, 2019) to find a sample-level embedding. Of note, this method uses supervised filters to find groups of cells related to COVID-19 mortality before sample-level analysis. The latter methods (Kuchroo et al, 2022; Ravindra et al, 2020; Zhang et al, 2022) require labels of patients for their analysis and cannot be used in the unsupervised analysis (clustering or trajectory inference) of patient-level single-cell experiments.

In this study, we introduce PILOT (Patient level distance with Optimal Transport), which explores optimal transport for sample-based analysis of multi-scale single cell or pathomics data. First, we introduce PILOT's framework and its main methodological features. Next, we perform a benchmarking study to compare PILOT and competing approaches on their performance in 12 public single-cell and pathomics data sets. This indicated favorable results of PILOT in both clustering and trajectory prediction problems. Finally, we showcase PILOT's features by interpreting trajectory predictions on a single cell data set of samples with myocardial infarction and a pathomics data set on samples with kidney disease; and clustering analysis of a single cell data with pancreatic adenocarcinoma patients.

# Results

## Patient level distance with Optimal Transport (PILOT)

Upon diseases, tissues undergo cellular and tissue remodeling changes. For example, in myocardial infarction cardiomyocytes acquire injury cell states, immune cells migrate to injured tissue and fibrosis or scarring takes place to compensate tissue loss due to necrosis (Zhang et al, 2022). We hypothesize therefore that changes in cellular composition are hallmarks of disease progression. We propose PILOT—Patient level distance with Optimal Transport (OT)—to perform sample level unsupervised analysis of multi-scale single cell or pathomics data. PILOT is based on three major modules: (1) a module using optimal transport (Peyré and Cuturi, 2019) to find similarities between samples; (2) an unsupervised analysis part to find sample level trajectories or clusters; and (3) an interpretation module to delineate cellular, molecular or structural features associated with the predicted clusters or trajectories (Fig. 1). In the first module, PILOT models each sample/patient as a distribution of cells into clusters. Here, a cluster represents a group of cells, which can be annotated to a particular cell type or a cell state. This distribution is encoded as a matrix $P = \{p_{lk}\}^{NxK}$ where $p_{lk}$ indicates the probability to find a cell in cluster $k$ at patient $l$. It then uses optimal transport to find a transport plan ($T$) by moving masses of cluster distributions between a pair of sample $l$ and $q$. PILOT explores the assumption that some cellular changes, i.e., the change of healthy cardiomyocytes to injured cardiomyocyte state are short-term events in disease progression, while tissue remodeling (e.g., replacement of lost cardiomyocytes by fibroblasts due to scarring process) is indicative of a long term event in disease progression (Fig. EV1). Therefore, it defines a cost matrix ($C$) so that transporting masses between similar cell clusters (healthy vs. injured cardiomyocytes) have a lower cost than transporting masses between distinct cell clusters (cardiomyocytes vs. fibroblasts). $C$ is estimated by the similarity between the centroids of clusters. Finally, the Wasserstein distance is estimated as the total cost associated with the optimal transport plan ($T$; Eqs. (1) and (2) in Fig. 1). By repeating this for all pairs of samples, we obtain a distance matrix $W$ defined over all samples. Next, in the unsupervised analysis module, PILOT uses $W$ to: (1) infer sample-level disease progression trajectories by the use of diffusion maps (Coifman et al, 2006) followed by a path inference algorithm (Albergante et al, 2020) or (2) find clusters of samples with a graph-based algorithm (Traag et al, 2019). In the interpretation module, PILOT uses statistical methods to find features associated with estimated trajectories or clustering. For trajectories, PILOT uses robust non-linear regression models (Huber, 1965) to find associations of cell clusters, genes, or morphological features associated with the estimated disease progression. The model fit (using a $F$-statistic test) indicates the significance of the detected associations. In the case of genes and morphological features, the test can be conditioned on a cluster/cell type. We use a Wald test that tests if the model fit of a gene in a cell cluster differs from the model when considering all other cell clusters (Van den Berge et al, 2020). That is, the gene expression of the cluster is associated with time and differs from the expression of other clusters. For the clustering problem, we use a non-parametric Welch's t-test to evaluate if a cell cluster proportions changes between two clusters of samples. For cell clusters with significant changes, PILOT allows the finding of cell cluster-specific expression changes for samples in two clusters by using a limma-based differential expression test (Ritchie et al, 2015). Thus PILOT represents the unique approach to perform sample level detection of unknown patient trajectories and clusters, while providing statistical models for interpretation of cellular, molecular, and morphometric features associated with disease progression or clusters.

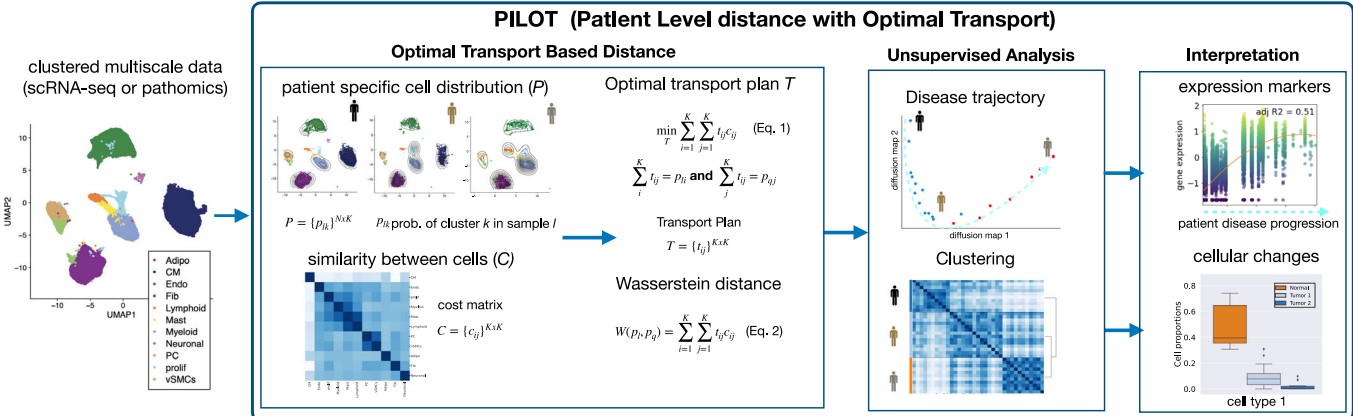

**Figure 1. PILOT schematic.**

PILOT receives clustered and integrated scRNA-seq or pathomics data as input. In the optimal transport module, PILOT estimates patient-specific cluster distribution $P$ and a cost function $C$ between clusters. These are used as input for an optimal transport algorithm (Peyré and Cuturi, 2019), which estimates the transport plan $T$ for a pair of cluster distributions (Eq. (1)). This optimal transport plan can be used to estimate the distance between all pairs of samples (Eq. (2)) providing a sample-specific distance matrix $W$. This is used as input for the unsupervised module, which detects sample specific trajectory and clustering. Finally, the interpretation module provides statistical models to characterize cellular, molecular, or morphological features associated with sample trajectories or clusters.

## Evaluation of patient-level clustering and trajectory analysis

We compare the results of PILOT and PhEMD (Chen et al, 2020) and baseline methods in the recovery of known patient groups of eight single-cell RNA data sets (PBMCs from systemic lupus erythematodes (Perez et al, 2022), pancreatic ductal adenocarcinoma [PADC] (Peng et al, 2019), acute myocardial infarction (Kuppe et al, 2020), PBMC of patients with COVID-19 infection (Stephenson et al, 2021), kidney injury (Lake et al, 2023), lung cancer (Sikkema et al, 2023), follicular lymphoma (Han et al, 2022) and diabetes (Hrovatin et al, 2022)) and four pathomics data sets with morphological features of glomeruli and tubules from two distinct disease cohorts (kidney IgA nephropathy of the VALIGA study [IgAN] (Coppo et al, 2014) and kidney biopsies of the Aachen cohort [AC] (Hölscher et al, 2023)). For trajectory analysis, we also evaluate the use of cell proportions followed by a PHATE embedding (proportions+PHATE), as this approach is equivalent to the analysis reported in Multiscale-PHATE (Kuchroo et al, 2022)[1]. For most of these data, we use disease status (control and case) as labels for the clustering task. For the Kidney IgAN pathomics data, patients/donors were labeled by their estimated glomerular filtration rate (eGFR; normal, reduced, and low); for the COVID-19 samples were labeled as control, mild and severe cases; for the scRNA-seq kidney injury data samples were grouped as normal, chronic kidney disease and acute kidney failure. The lung and pancreas cell atlas have controls and distinct disease types (4 for lung and 3 for pancreas). These are the largest publicly available single cell and pathomics data with up to millions of cells and structures and measured in hundreds of patient samples (Table 1). We also evaluate two baseline methods: pseudo-bulk libraries per sample by using state-of-art RNA-seq pipelines (Witten, 2011) or clustering directly the cell proportion matrices.

For scRNA-seq data, we used the data as provided in the original manuscript as input for PILOT. For pathomics data, structure segmentation was performed with FLASH followed by graph-based clustering as described before (Hölscher et al, 2023). We evaluate these methods in distinct scenarios. In the clustering evaluation problem, we evaluate the methods regarding their performance in clustering the data using a graph-based clustering (Traag et al, 2019). The number of clusters was the same as the number of true classes in the data. The accuracy of the methods is evaluated with the well-known external clustering index, the adjusted Rand index (ARI) (Rand, 1971). Next, we use the Silhouette index (Rousseeuw, 1987) to evaluate how well separated samples with the same labels are, according to the distance estimated by each method (distance evaluation). This evaluation does not require a clustering of samples. Finally, we evaluated all methods regarding prediction of disease trajectories, i.e., the relation of sampling ordering with the class labels. For this purpose, we use the Area Under the Recall Precision Curve statistic (AUCPR; Trajectory Evaluation). For each of these three scenarios, we use the Friedman–Nemenyi test to contrast the performance of the methods in distinct data sets. This non-parametric test uses the rank of each algorithm per data set to find the algorithms outperforming competing approaches (Demšar, 2006).

Regarding the clustering problem. PILOT obtained the highest average ARI score and its ranking was significantly superior than all competing methods (Fig. 2A, $p$-value <0.05; Friedman–Nemenyi test, and Appendix Table S1). Concerning the distance evaluation, PILOT had the highest ranking among all evaluated methods (Fig. 2B) and outperformed PhEMD ($p$-value <0.05; Friedman–Nemenyi test). Finally, regarding the trajectory analysis, we observe that PILOT has the highest mean rank when considering AUCPR values and outperforms PhEMD, Pseudo-Buk and proportions-PHATE (Fig. 2C, $p$-value <0.05; Friedman–Nemenyi test). While there is no statistical difference between the performance of PILOT and the proportions, PILOT had a higher AUCPR in eight of the 12 evaluated methods and was only seconded by the proportion in one single data set (Appendix

---

[1]We could not replicate the analysis in Multiscale-PHATE due to the lack of code.

**Table 1.** Characteristics of data sets used for benchmarking.

| Data | Type | #Cells/ Structures | #Samples | #Cell_types | #Classes |
|---|---|---|---|---|---|
| Lupus PBMC (Perez et al, 2022) | scRNA-seq | 1,263,676 | 261 | 11 | 2 |
| COVID-19 PBMC (Ren et al, 2021) | scRNA-seq | 993,171 | 151 | 10 | 3 |
| Lung (Sikkema et al, 2023) | scRNA-seq | 941,504 | 165 | 33 | 5 |
| Diabetes (Hrovatin et al, 2022) | scRNA-seq | 264,235 | 52 | 13 | 4 |
| Follicular lymphoma (Han et al, 2022) | scRNA-seq | 137,147 | 23 | 18 | 2 |
| Myocardial Infraction (Kuppe et al, 2020) | scRNA-seq | 115,517 | 20 | 11/33 | 2 |
| Kidney (Lake et al, 2023) | scRNA-seq | 76,020 | 36 | 57 | 3 |
| Pancreas (PDAC) (Hrovatin et al, 2022) | scRNA-seq | 57,530 | 35 | 10 | 2 |
| Kidney IgAN (Tubule) (Hölscher et al, 2023) | Pathomics | 64,493 | 634 | 15 | 3 |
| Kidney AC (Tubule) (Hölscher et al, 2023) | Pathomics | 56,998 | 57 | 7 | 2 |
| Kidney IgAN (Glomeruli) (Hölscher et al, 2023) | Pathomics | 24,227 | 634 | 14 | 3 |
| Kidney AC (Glomeruli) (Hölscher et al, 2023) | Pathomics | 4731 | 57 | 28 | 2 |

Table S1). In addition, for data sets with more than 2 classes (COVID-19 PBMC, Lung, Diabetes, Kidney/scRNA, and Kidney IgAN), we created an ordered variable, whose value increases with disease severity (control = 1, mild = 2, severe = 3). Note that for lung, we have classified all carcinomas (SCLC, LA, NSCLC) as severe and chronic obstructive pulmonary disease (COPD) as mild. For diabetes, we classified type I and II diabetes as severe and endocrine pancreas disorder as mild. This is motivated by the lack of guidelines to discriminate the severity of types of diabetes or lung carcinomas. We then measure the Spearman Correlation between the estimated disease progression scores and the ordered values (control = 1, mild = 2, severe = 3). This shows similar results as with the AUCPR evaluation, where PILOT is ranked first and having similar results with proportions (Fig. 2D, $p$-value < 0.05; Friedman–Nemenyi test and Appendix Table S2). To further verify the value of disease progression score, we compare the Spearman Correlation when shuffling scores for control, mild, and severe cases. The use of the ordering "Control < Mild < Severe" yields the highest SC values in all evaluated data sets (Fig. EV2A).

Due to the large size of data sets, computing time is also a relevant aspect. In the Lupus PBMC data with millions of cells, PILOT required 58.12 s versus 9.03 s of the simple pseudo-bulk and 16.50 min of PhEMD[2]. These indicate that PILOT and some competing methods can be efficiently run in large data with millions of cells or tissue structures.

We observe that diffusion maps estimated with PILOT recovered trajectory-like structures in all analyzed data sets (Fig. 2E,F; Appendix Figs. S2 and S3). Also, disease progression score is related with the severity of diseases in the data sets (Fig. 2E,F; Appendix Fig. S4). For example, in the lung cell atlas normal samples were at the beginning of the trajectory. These were followed by samples associated with chronic lung disease (COPD), while acute carcinoma samples (SCLC, LA, NSCLC) accumulated in the end of the trajectory. For Kidney IgAN (Tubule), we observe the trajectory order samples regarding normal, low, or reduced glomerular filtration rate. These results

are also reflected in cumulative distributions of control, mild and severe labels overestimated disease progression (Fig. EV2B) and support that PILOT can sort controls, moderate, and severe disease cases.

A point that has been poorly addressed so far is the impact of technical artifacts in single-cell disease atlases. To address this, PILOT explores statistical tests to associate estimated clustering or trajectories with biological or technical features describing samples. In the context of clustering, we observe that the status variable (class labels) is significantly associated with the clusters in all datasets (see Appendix Table S3). This supports the predictive performance of PILOT. Interestingly, we also observed that for the COVID-19 PBMC scRNA-seq data, a variable describing the city of origin of the sample had a stronger association with the clustering than the status variable. Similarly, the study of the origin of the samples in the Lung Single-Cell Atlas was also associated with PILOT's clusters. These results indicate the potential presence of technical artifacts in these datasets and demonstrate how PILOT's sample-level analysis can support the quality check of disease single-cell atlases.

PILOT requires no definition of parameters. However, it assumes that single-cell experiments have been previously pre-processed and clustered. Currently, PILOT adopts the same clustering and integration strategy as in the manuscript describing the data, as cells are labeled and this helps the interpretation of results. To investigate if clustering can impact PILOT, we change the resolution parameter of the Leiden Clustering algorithm for selected data sets. We observe that this parameter is not critical for the performance of PILOT in clustering samples (Appendix Fig. S5A,B). We also evaluated the impact of distinct integration methods (harmony (Korsunsky et al, 2019), bknn (Polanski et al, 2020), and scanorama (Hie et al, 2019)). Results indicate there is limited impact of these methods in PILOT's predictive performance (Appendix Fig. S5C–E). Finally, we investigated the effect on the similarity/distance measure used to estimate the cost function ($C$) for optimal transport. We observe that the Cosine similarity is ranked first and is the only metric outperforming other approaches in most scenarios ((Euclidean, Manhattan, and Chebyshev; Appendix Fig. S5E–G). This supports the use of the Cosine similarity by PILOT.

---

[2]This computing time of PhEMD also includes clustering of single cells, as this is a required step in PhEMD framework

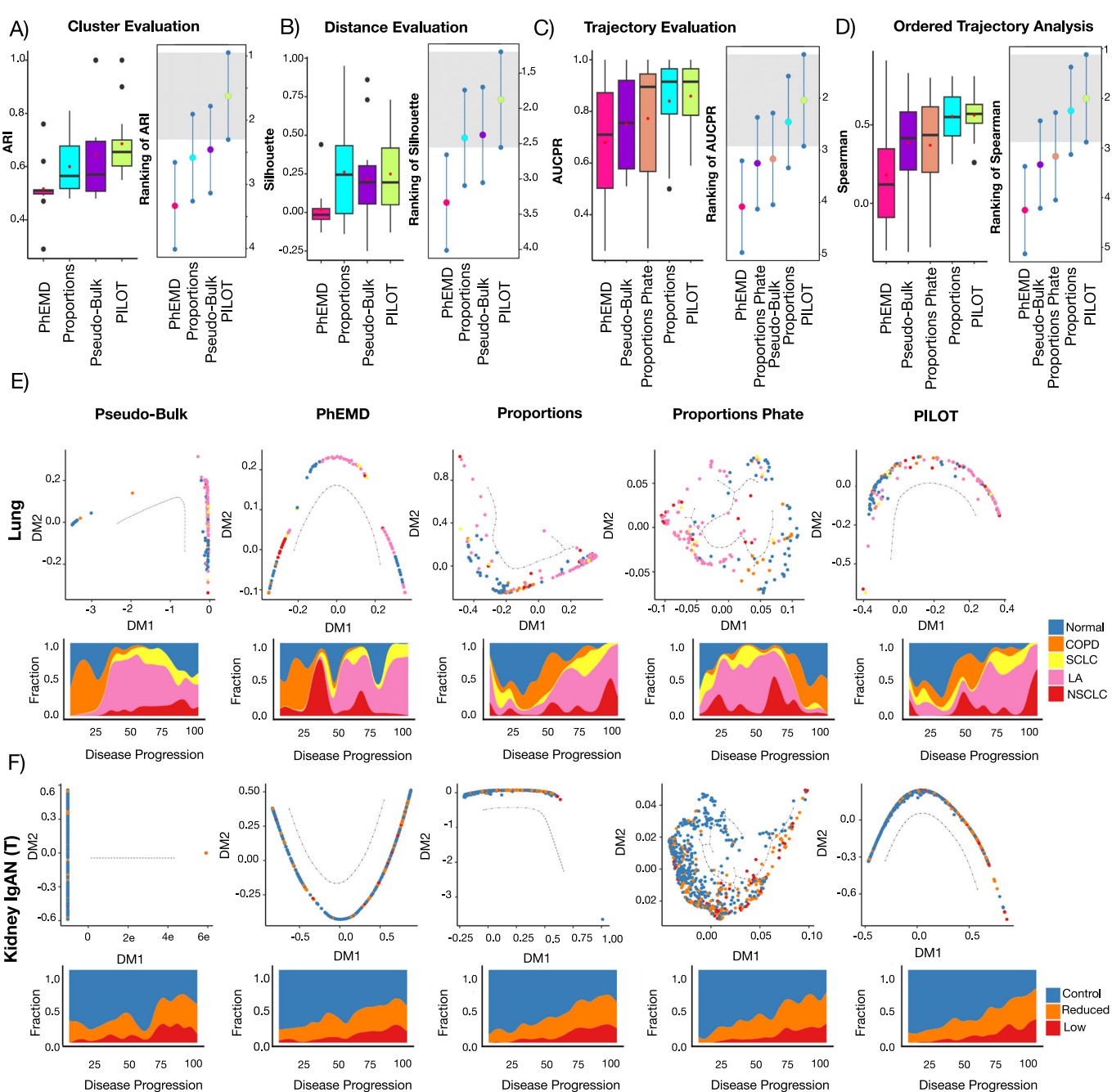

**Figure 2. Benchmarking of Patient Level Clusters and Trajectories.**

(**A**) Box plot with ARI values (y-axis) for distinct evaluated methods (x-axis) (left) and ranking distributions (x-axis) based on the Friedman–Nemenyi test for distinct methods (right) ($n = 12$). Methods with average ranking in the gray area are top performers. (**B–D**) E quivalent to (**A**) for Silhouette, AUCPR, and Spearman rank correlation statistics. In box plots (**A–D**), Boxes represent interquartile range (IQR) between the 25th and 75th percentiles (Q1 and Q3) and the line inside the box represents the median value. The center point denotes the mean, while the whiskers extend to the minimum and maximum values. (**E**) Diffusion maps (top) and fraction of patient labels vs. pseudotime (bottom) on lung cell atlas data set ($n = 165$) for distinct algorithms. For this data, labels COPD, SCLC, LA, and NSCLC correspond to Chronic Obstructive Pulmonary Disease, Squamous Cell Lung Carcinoma, Lung Adenocarcinoma, and Non Small Cell Lung Carcinoma groups, respectively. (**F**) Diffusion maps (top) and fraction of patient labels vs. pseudotime (bottom) for Kidney IgAN (Tubule, ($n = 634$)). Source data are available online for this figure.

## PILOT trajectories detect events associated to cardiac remodelling in myocardial infarction

Next, we analyze the inferred trajectory from normal heart tissue towards ischemic zone (IZ) tissue in acute myocardial infarction. The trajectory reflected the known annotation of samples with the exception of one case in the decision boundary between controls/IZ samples (Fig. 3A). We use non-linear regression methods to find cellular and molecular changes associated with the disease progression score. Major cellular changes, as indicated by the highest adjusted $R^2$ values, include a quadratic increase of SPP1

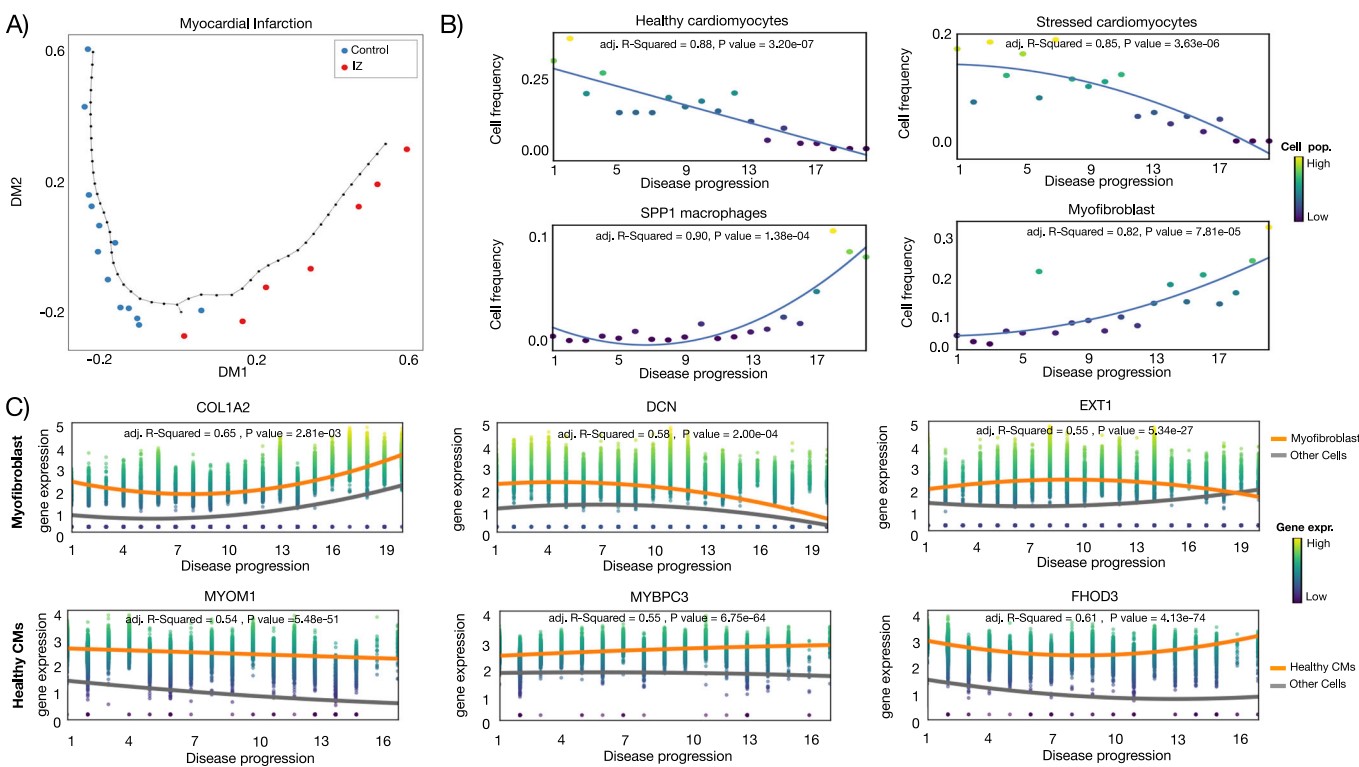

**Figure 3. Sample-level analysis of myocardial infarction.**

(A) Diffusion map and disease progression trajectory in myocardial infarction ($n = 20$). (B) Fraction (y-axis) of top four cells (highest R2) vs. disease progression (x-axis). (C) Expression (y-axis) of selected genes with significant changes in disease progression (x-axis) for healthy cardiomyocytes and myofibroblast cells. Each circle in the plot represents an individual cell, and the color intensity corresponds to the magnitude of gene expression for the respective gene. The gray line shows the model estimation in other cell clusters. For (B, C), we use the Wald test to evaluate the difference in the fitted model in the target cells vs. the model fitted for background cells. Source data are available online for this figure.

positive macrophages and myofibroblasts, a quadratic decrease of healthy cardiomyocytes followed by a later and smoother decrease of stressed cardiomyocytes during disease progression (Figs. 3B and EV3). These patterns are in accordance with the cellular changes expected from early myocardial infarction, which include damage of myogenic tissue (less cardiomyocytes) followed by inflammation (increase in immune cells) and fibrosis (increase in fibroblast cells) (Zhang et al, 2022). This example also confirms the power of PILOT predicting trajectories to find continuous changes related to tissue remodeling.

One open point from our benchmarking analysis was the fact that clustering resolution did not affect the performance of PILOT in sample clustering. To investigate if this impacts the trajectory analysis and the interpretation of results, we execute PILOT with a broader clustering (11 annotated clusters), which is obtained by joining the more granular 33 clusters that were used before. We observe that while global changes are equivalent (decrease of cardiomyocytes and increase of fibroblasts and myeloid cells), this analysis loses a lot of important nuances (Fig. EV3B). For example, it does not reflect the fact that damaged cardiomyocyte populations have a slower decay than healthy cardiomyocytes or that the increase in Fibro Scara5+ progenitor cells precedes the increase in the differentiated myofibroblast cells. These results support the advantage of using coarse clustering and annotated cell clusters in the interpretation of trajectories.

Next, we perform an explorative analysis on the genes selected by PILOT using the Wald test. Gene ontology enrichment analysis (Appendix Fig. S6) indicates that PILOT identifies genes related to extracellular matrix remodeling to be induced by myofibroblasts (Appendix Fig. S7) and genes associated with muscle and muscle function in healthy cardiomyocytes (Appendix Fig. S8). Myofibroblast genes include COL1A2, which has an overall quadratic increase in gene expression upon disease progression (Fig. 3C). Decorin (DCN) and exostosin-1 (EXT1) are examples of genes with decreased expression upon late stages of disease progression. Decorin is a fibroblast-specific gene with antifibrotic properties due to TGFB signaling inhibition (Baghy et al, 2011; Isaka et al, 1996). Exostosin-1 has been associated with early formation of collagen fibers (Hill et al, 2022). We observe several genes related to muscle function and organization (MYBPC3, FHOD3, MYOM1) to have expression changes in healthy cardiomyocytes. MYBPC3, which is a gene with a slight but consistent increase in expression over the trajectory, has been shown to support cardiomyocyte proliferation (Jiang et al, 2015) and is related to hypertrophic cardiomyopathy (Hershberger et al, 2010). MYOM1 and FHOD3 are gene important in the regulation actin filaments and sarcomeres structures (Lamber et al, 2022; Taniguchi et al, 2009). FHOD3 has a decrease in expression similarly at the middle of the trajectory indicating a potential role which would have been missed if not analyzed within

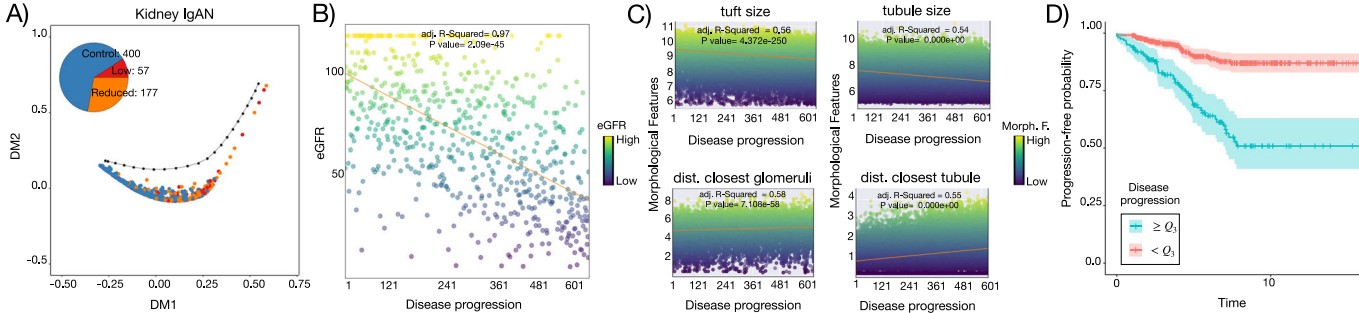

**Figure 4.  Sample level analysis of kidney IgAN.**

(A) Diffusion map and disease progression trajectory in kidney IgAN ($n = 634$). Pie chart indicates the number of samples in terms of their status. (B) eGFR values (y-axis) vs disease progression (x-axis). Each circle in the plot represents an individual structure, and the color intensity corresponds to the magnitude of morphological feature for the respective structure ($n = 634$). (C) Morphometric values (y-axis) of kidney structures vs disease progression (x-axis). P-value of (B, C) were estimated with the F-statistic test. (D) Kaplan–Meier plot with the time to kidney failure from patients within the upper quartile (high progression score) vs. other samples (low progression score) ($n = 634$). Source data are available online for this figure.

temporal context. Altogether, these results support how PILOT can detect molecular and cellular changes at distinct stages of myocardial infarction.

### PILOT trajectories in pathomics data

We performed a trajectory analysis of the pathomics data of kidney IgAN biopsies. As distinct morphological features are used to describe tubules and glomeruli, it is not possible to cluster these structures together, so these data are independently analyzed as in (Hölscher et al, 2023). We combine the two disease progression scores with their sum, which yields similar or higher AUCPR scores than the trajectories considering one structure at a time (Figs. 4A and EV4). This trajectory is associated with a linear decrease in eGFR (Fig. 4B), which is the current surrogate used in clinical practice to estimate kidney function. Regarding morphometric features, PILOT detects a significant linear decrease of glomerular tuft sizes and tubule sizes; and a linear increase in distance between glomeruli and tubules as significant characteristics of disease progression (Fig. 4C; Appendix Figs. S9 and S10). These are indicative of interstitial fibrosis, tubular atrophy, and glomerulo-sclerosis, which are all hallmarks of kidney function decline. Finally, we made use of a prognostic variable from the kidney IgAN biopsies, which indicates if patients progressed to kidney failure. PILOT progression score is more associated with kidney failure (p-value of 2.4e−11; likelihood ratio test) than a multivariate model combining all morphometric variables (p-value of 7e−05; likelihood ratio test) or the use of individual morphometric variables (Appendix Table S4). Indeed, samples with higher disease progression scores (top 75 quartiles) have a higher chance of kidney failure than other samples (Fig. 4D). Altogether, these results reinforce the power of PILOT in discriminating disease outcomes from pathomics data.

### PILOT detects subgroups of pancreatic adenocarcinoma patients

As a case study for using PILOT to find sub-clusters of samples (Appendix Fig. S1), we performed graph-based clustering (Traag et al, 2019) of the pancreatic ductal adenocarcinoma (PDAC) at distinct resolutions. We employed the Silhouette Score (Rousseeuw, 1987) to determine the optimal number of clusters for varying the

resolutions. The optimal resolution (0.3) had three clusters: the first includes 11 control samples, and the remaining two are associated with PDAC samples, comprising 14 samples classified as Tumor 1 and 10 samples as Tumor 2 (Fig. 5A). At the cell proportion level, these two PDAC clusters differed by their amount of ductal cell 2, (higher in Tumor 2 samples), stellate cells and ductal cell 1 (higher in Tumor 1 samples; p-value < 0.05; Fig. 5B,C). Ductal cell 2, which is more prevalent in both Tumor 1 and 2 samples than in controls, was reported in the original study (Peng et al, 2019) to be associated malignant cells, due to the higher occurrence of PADC-associated copy number variations. Differential expression and GO analysis contrasting ductal cell 2 expression for Tumor 2 vs. Tumor 1 samples indicate the up-regulation of the hypoxia related "HIF-1 signalling pathway" (Fig. 5D,E). Regarding stellate cells, genes upregulated in Tumor 2 samples have characteristic of fibrosis as Collagen and matrix-associated genes, while downregulated genes are associated with pancreas functions (Fig. 5F,G). These indicate that despite the decrease of stelate cell populations, they acquire a fibrotic signature. Fibrosis is known to further potentiates hypoxia, which can trigger metastasis of PDACs (Shah et al, 2020). This exploratory analysis is an example of how PILOT can be used to characterize potentially clinically relevant findings in patient single-cell data sets.

## Discussion

The technological improvements in single-cell genomics and digital pathology are providing us with clinically rich and large-sized data describing cellular and morphological changes in diseases. We present here PILOT—a computational pipeline for the detection and feature characterization of disease trajectories from single-cell genomics or pathomics data. By using a comprehensive benchmark with twelve data sets, we show that PILOT is superior to the competing approaches in both clustering and disease trajectory prediction. Of note, all previous work (Chen et al, 2020; Flores et al, 2023; Kuchroo et al, 2022; Zhang et al, 2022) based their analysis of the exploratory analysis of individual data sets, and did not performed any benchmarking. Another important aspect is the

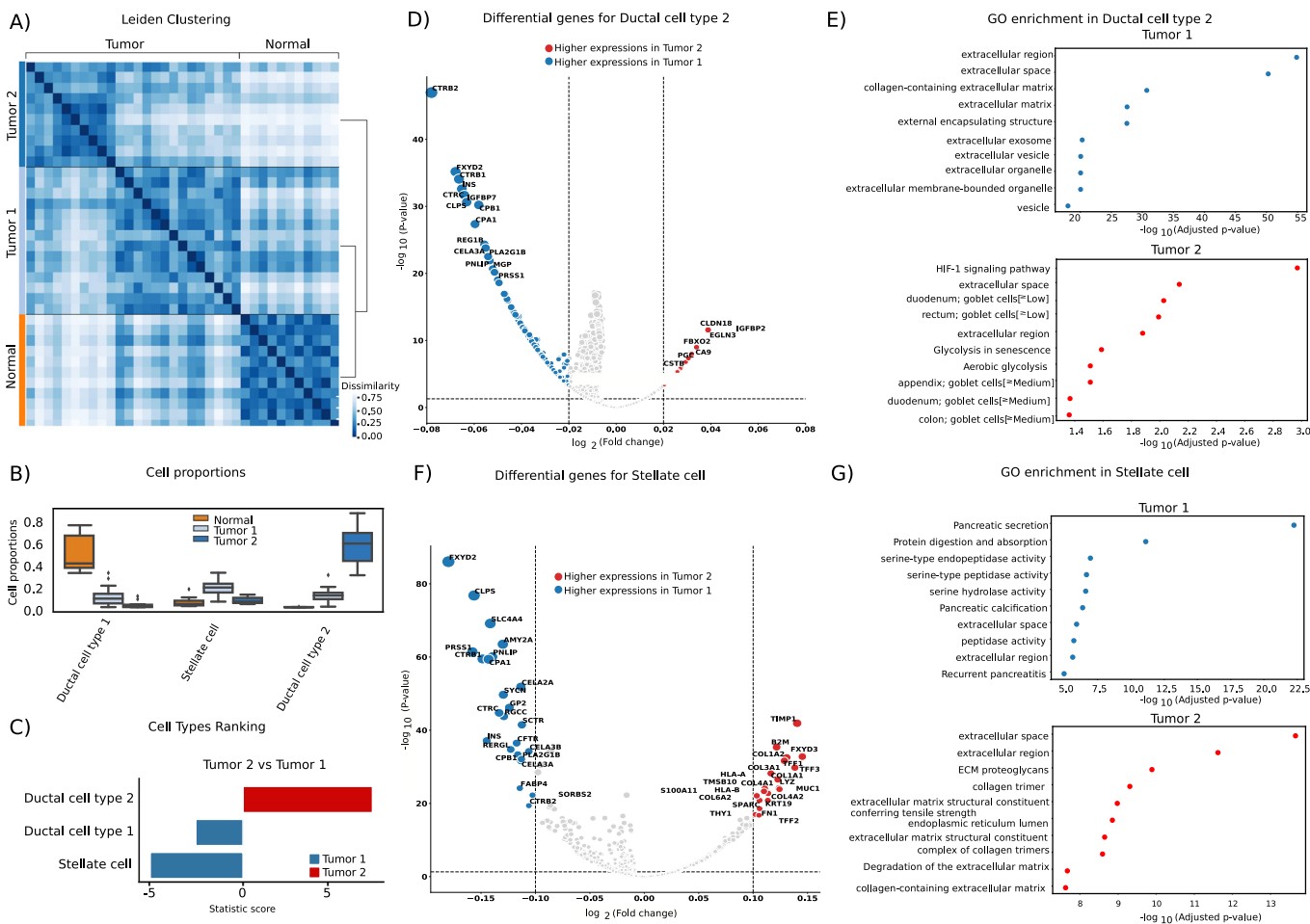

**Figure 5. Clustering analysis of pancreas ductal adenocarcinoma (PDAC, $n = 35$).**

(A) Heatmap with Leiden clustering results, which supports the existence of two subclasses of PDAC patients. (B) Boxplot with cell cluster proportion (y-axis) distribution per sample cluster ($n = 35$). Boxes represent the interquartile range (IQR) between the 25th and 75th percentiles (Q1 and Q3), including the median (the lines inside). The whiskers extend to the minimum and maximum values. (C) T-statistic (x-axis) of cell clusters with significant changes in Tumor 1 vs. Tumor 2 samples. (D) Differential expression and (E) Gene Ontology Enrichment analysis contrasting Tumor 2 vs. Tumor 1 samples in ductal cell cluster 2. (F, G) Same as (D, E) for stellate cell. P-values shown in (D) and (F) were estimated with empirical Bayes moderated t-test from limma. Gene enrichment analysis for (E) and (G) ($n = 10$) is performed with g:profiler (Fisher's Exact test). Source data are available online for this figure.

interpretation of predictions. PILOT uses a robust non-linear regression model or statistical tests, which can predict cellular populations, marker genes or morphological features associated with the disease progression or clusters. These different aspects make PILOT a unique framework for sample-level analysis of multi-scale single cell or pathomics data.

We revisited the analysis of the largest single-cell genomics data on myocardial infarction (Kuppe et al, 2020), where PILOT could successfully predict a trajectory from controls toward ischemic samples. This allowed us to find non-linear changes in cell composition during cardiac remodeling, i.e., quadratic decrease of healthy cardiomyocytes and quadratic increase of myofibroblast and macrophages cells. Similarly, PILOT dissected gene expression programs associated with these changes, such as non-linear increase in expression of extracellular matrix-related genes in myofibroblasts and increase of genes associated with cardiomyopathy in healthy cardiomyocytes. We also evaluated the power of PILOT in inferring a disease trajectory of pathomics data from

patients with kidney IgAN. We show that disease progression as estimated by PILOT is related to eGFR and provides a better predictor for future kidney failure than the use of the morphometric features alone or together. This highlights the power of PILOT in finding non-linear manifolds to model disease progression. In the case study consisting of clustering different samples, PILOT characterized a population of patients with pancreatic adenocarcinoma cells, who display hypoxia and fibrotic expression signatures.

An important aspect of PILOT (and competing methods) is the assumption that tissue samples were collected using an uniform approach, i.e., similar tissue areas and the same technology. The analysis of data from distinct technologies (single nucleus vs. single cell) or distinct single cell isolation methods (droplet based vs. well based) is still an open challenge for future research. While PILOT does not explicitly correct for such potential technical artefacts, it provides statistical tests to evaluate if technical factors might influence the predictions. A systematic analysis in clustering

predictions indicates the presence of technical artifacts in a lung (Salcher et al, 2022; Sikkema et al, 2023) and COVID-19 (Ren et al, 2021) single-cell atlases. While this was discussed in the lung cell atlas study (Sikkema et al, 2023), it is unclear how this affects comparison of disease groups of either data set. However, dimension reduction methods and clustering of the samples are classical approaches to perform quality checks in bulk transcriptomics. PILOT's sample-level analysis represents an alternative of such analysis for multi-scale data, such as single-cell disease atlases.

It is also important to stress that some of the competing approaches, such as SCANCell (Zhang et al, 2022), MOFAcellulaR (Flores et al, 2023), or the work by Cain et al (Cain et al, 2023), also present alternative approaches to interpretation of cellular changes from sample cohorts. Particularly interesting are methods to infer cellular communities associated with sample groups (Cain et al, 2023; Zhang et al, 2022). Molecular studies of diseases are increasingly based on multi-modal measurements; gene expression, protein abundances, chromatin accessibility, and histology images at either single cell and/or spatial level (Marx, 2022). Future work of multi-scale level analysis will require methods considering the multi-modal and spatial nature of these data into account.

# Methods

## PILOT

We formalize the main steps of PILOT[3]. In analyzing single-cell experiments, assume there are $L$ single-cell matrices $\mathcal{X} = \{\mathbf{X}_1, ..., \mathbf{X}_l, ..., \mathbf{X}_L\}$, where $\mathbf{X}_l \in \{\mathbb{R}\}^{N_l \times D}$, where $N_l$ represents the number of cells in sample $l$ and $D$ is the number of features (genes), which is common between all experiments. In practice, PILOT receives as input a single matrix $\mathbf{X}^I \in \{\mathbb{R}\}^{N \times D}$ after integration of the matrices in $\mathcal{X}$, where $N$ represents the total number of cells. To keep the sample information, we define a vector indicating the sample identity of cell $i$, i.e., $s = \{s_1, ..., s_N\}$, where $s_i \in \{1, ..., L\}$ indicates which sample (patient) the cells belongs to. A usual representation/sumarization of single cell experiments is to group cells with a clustering algorithm. This can be represented by a vector $y = y_1, ..., y_N$, where $y_i \in \{1, ..., K\}$ indicates the group of cell $i$.

Our main problem is to estimate the distance between scRNA-seq data measured over two distinct samples (patients), where a scRNA-seq is represented as a set of clustered cells. Here we explore the concepts of the optimal transport-based Wasserstein method (Bonneel et al, 2011) to compare two samples by representing samples as distributions of cells. First, we defined $p_{lk}$ as the probability of having a cluster $k$ for patient $l$:

$$p_{lk} = P(y = k | s = l). \tag{1}$$

This can be used to define a probability distribution vector $p_l = (p_{l1}, ..., p_{lK})$.

From these, we use optimal transport to find the optimal transport plan $T = \{t_{ij}\}^{K \times K}$ between two distributions $p_l$ and

---

[3]For simplicity, we focus here on single-cell data, but the same formalism applies to pathomics data

$p_q$ describing samples $l$ and $q$ by considering that there is a cost $c_{ij}$ associated with moving some mass between cluster $i$ to cluster $j$:

$$\min_T \sum_{i=1}^{K} \sum_{j=1}^{K} t_{ij} c_{ij}, \tag{2}$$

such that $t_{ij} \geq 0, \sum_i^K t_{ij} = p_{lj}$ and $\sum_j^K t_{ij} = p_{iq}$ and $\sum_{i=1}^{K} \sum_{j=1}^{K} t_{ij} = 1$.

For a given optimal transport plan $T$, the Wasserstein distance ($W$) is calculated as:

$$W(p_l, p_q) = \sum_{i=1}^{K} \sum_{j=1}^{K} t_{ij} c_{ij}. \tag{3}$$

The optimal transport matrix $T$ can be estimated using a minimum flow cost algorithm. PILOT is based on the emd2 function from POT library (Flamary et al, 2021), which implements the best solver described in (Bonneel et al, 2011). By estimating the Wasserstein distance between all pairs of samples, we obtain a distance matrix $W$ between all samples.

This framework, which is also denoted Earth Moving Distance (Rubner et al, 2000), is also used in PhEMD. PILOT, however, addresses important issues which are crucial in the noisy and large nature of single cell and pathomics data. These are namely: (1) how to obtain robust estimates of probability distributions due to potential cell content bias and low cell coverage of individual samples; (2) how to estimate the cost matrix $C$ on the large matrices $\mathbf{X}^T$; and (3) by offering statistical models to select features associated with disease progression. These three points are described below.

## Robust estimation of sample probability distributions

The coverage of cells per cluster can vary across distinct samples. This effect is potentially higher in diseased samples, due to their lower cell viability. Let $z_l = (z_{l1}, ... z_{li}, ..., z_{lK})$ be a vector, where $z_{lk}$ is the number of cells in a cluster $k$ and sample $l$. We define the following hierarchical model:

$$\begin{aligned} z_l &\sim Multi(N_l, \Theta) \\ \Theta &\sim Dir(\alpha) \end{aligned} \tag{4}$$

where $N_l$ is the number of cells in the sample $l$, $\Theta$ is a random variable representing distributions $p$ and $\alpha$ is the hyper-parameter of a Dirichlet distribution. The posterior distribution can be re-written as

$$p(\Theta | z_i, \alpha) \sim Dir(z_i + \alpha) \tag{5}$$

We can use this to obtain maximum a posteriori estimates of $\Theta$, i.e.,

$$\hat{p}_{lk} = \frac{n_{kl + \alpha_k}}{n_l + \sum_{i=1}^{K} \alpha_i}. \tag{6}$$

Here, we use the following parametrization of the prior $\alpha_k = N_k / N * c$, where $c$ is set as 0.1 as default. This prior adds "pseudo cell counts", which are weighted by the cell distribution of all samples.

This prior mitigates the fact rare cells might not be observed in samples with low cell coverage.

## Estimation of cost matrix

The optimal transport also considers the cost $c_{ij}$ of transporting a distribution mass from a cluster $i$ to a cluster $j$. For this, we generate a median representation for each cluster (in the PCA space) and estimate the cosine distance between the median value of clusters $i$ and $j$ as the cost to transport masses. The cosine distance between the median of cluster $i$ ($M_i$) and cluster $j$ ($M_j$) is defined as:

$$c_{ij} = 1 - \frac{M_i \cdot M_j}{\|M_i\|_2 \|M_j\|_2} \tag{7}$$

Of note, PhEMD uses centroids and Euclidean distances in non-linear embedding to estimate the cost matrices. The non-linear embedding assumes a cellular continuum between all cells in the data, which is not present in whole tissue single cell or pathomics data. Moreover, median values reduce the effects of outliers.

## Clustering and disease trajectory estimation

The matrix $W$ (Eq. 3) provides the distance between all samples. Clustering analysis can be performed by providing $W$ as input to a Leiden clustering algorithm (Traag et al, 2019). PILOT also performs trajectory analysis by the use of diffusion maps (Coifman et al, 2005) followed by a trajectory estimation with EIPLGraph (Albergante et al, 2020). For this, we apply the Gaussian kernel to $W$ (Liu and Vinck, 2022) to construct the affinity matrix as $W^s$, i.e.,

$$W^s(w_i, w_j) = \exp\left(-\frac{(\|w_i - w_j\|)^2}{\varepsilon \rho(w_i)\rho(w_j)}\right), \tag{8}$$

where $\varepsilon$ and $\rho$ are the scale parameter and the bandwidth function (Berry and Harlim, 2016), respectively.

Next, we compute the transition matrix:

$$M = D^{-1}W^s, \tag{9}$$

where $D$ is a diagonal matrix with $d_{ii} = \sum_j W^s_{ij}$. Finally, we perform the spectral decomposition of matrix $M$ as:

$$M = D^{-1/2}V\Lambda V^T D^{1/2}, \tag{10}$$

where $\Lambda$ and $V$ are the eigenvalue and eigenvectors matrices. Ultimately we use eigenvectors with the highest values for obtaining diffusion maps. These are provided as input for EIPLGraph (Albergante et al, 2020), which infers a backbone of the trajectory. It also allows us to rank samples with a disease progression score $t = t_1, \ldots, t_L$, where $t_l$ is the ranking of the sample $l$. In our experiments, we only considered two highest eigenvectors. EIPLGraph also requires a root sample, which was visually selected to reflect parts of the trajectory with control samples.

## Identification of molecular and structural features associated with disease progression or sample groups

PILOT uses step-wise regression models to identify features (cellular abundances, gene expression or structural properties), whose values are consistently changing across disease trajectories. For this, it fits regression models with linear, quadratic, and linear-quadratic terms for each feature and uses statistical tests to determine the goodness of fit. We then report the most significant model for each feature.

Let $x_{ij}$ represent the expression of a feature (gene) $j$ in a cell $i$, and $p_i$ the pseudotime variable (associated with the pseudotime of its sample, $p_i = t_{s_i}$). In short, we fit the three regression models to the $j$th feature:

$$\begin{aligned} \hat{x}_{\cdot j} &= \hat{\mu}_j + \hat{\beta}_{1j}p, \\ \hat{x}_{\cdot j} &= \hat{\mu}_j + \hat{\beta}_{1j}p^2, \\ \hat{x}_{\cdot j} &= \hat{\mu}_j + \hat{\beta}_{1j}p + \hat{\beta}_{2j}p^2. \end{aligned} \tag{11}$$

where $\hat{\mu}$ is the models' intercept, and $\hat{\beta}$ are the models' coefficients. The models are ranked based on the coefficient of determination ($R^2$), and we only consider the model with the highest R-squared and at a significance level ($p$-values $< 0.05$). The same approach can be performed for distinct features, such as the proportion of cell clusters in the sample or morphological features.

In the case of gene expression, one is mostly interested in finding cell type (cluster) specific genes. Therefore, we only consider cells $i$ belonging to cluster $k$. Due to the sparsity of single-cell data, i.e., dropout events, we observed a single side tailed distributions of residuals. Therefore, to improve the robustness of our regression models, we use the Huber least square criterion (Eq. (12)) (Huber, 1992).

$$H(e_i) = \begin{cases} \frac{1}{2}e_i^2 & |e_i| <= \delta \\ \delta(|e_i| - \frac{1}{2}\delta) & |e_i| > \delta \end{cases} \tag{12}$$

where $e_i = x_{\cdot j} - \hat{x}_{\cdot j}$. The Huber regression lessens the effects of the outliers by using a term $\delta$ (default of 1.35). This defines residual values associated with outliers. For large $\delta$, the regression is equal to ridge regression.

Of note, the previous formulation requires a re-definition of the R-squared to consider the Huber regression penalization, that is:

$$\begin{aligned} R^2_{mod} &= 1 - \frac{SS_{res}}{SS_{tot}} \\ SS_{res} &= \sum_{j||x_{\cdot j}-\hat{x}_{\cdot j}|<=\delta} (x_{\cdot j} - \hat{x}_{\cdot j})^2 + \sum_{j||x_{\cdot j}-\hat{x}_{\cdot j}|>\delta} \delta\left(|x_{\cdot j} - \hat{x}_{\cdot j}| - \frac{1}{2}\delta\right) \\ SS_{tot} &= \sum_j (x_{\cdot j} - \overline{x}_{\cdot j})^2 \end{aligned} \tag{13}$$

Another relevant question is if the pattern of the feature (gene expression or morphometrics) in a given cluster over pseudotime is distinct from other clusters. To do this, we compare the expression patterns along the pseudotime between one cluster vs cells/structures from all other clusters (non-cluster). We utilized Wald statistics, using the linear combination of coefficients, to test whether there are differences between pair points of fitted curves (cluster vs. non-cluster). Subsequently, for each gene, we test $2 \times 3$

null hypotheses (between two models, each having at most three coefficients; see Eq. (11)) on the $n$ pseudotime points:

$$H_0 : C^T \hat{\beta} = 0$$
$$H_1 : C^T \hat{\beta} \neq 0 \tag{14}$$

where $C$ is the $(2 \times 3) \times n$ contrast matrix of interest and $\hat{\beta}$ are coefficients of two models. The distribution of the test statistic under the null hypothesis is:

$$W = \hat{\beta}^T C \left( C^T \hat{V}_{\hat{\beta}} C \right)^{-1} C^T \hat{\beta} \tag{15}$$

where $\hat{V}_{\hat{\beta}}$ is an estimator of the variance-covariance matrix of estimated coefficients. For large enough $n$, $W$ is distributed as $\chi^2$ with $r$ degrees of freedom d.f. for $n = 100$ (Harrell, 2001). To estimate the Wald score, we need to perform a eigendecomposition of the variance-covariance matrix ($\hat{V}_{\hat{\beta}}$). We consider all eigenvectors with eigenvalues larger than $1e^{-8}$. Statistical significance is estimated with the $\chi^2$ and the degree of freedom in the rank.

When clustering different samples, PILOT adopts distinct statistical tests. When calculating cell proportion changes, it uses the Welch t-test to compare changes in proportion between all pairs of groups. Regarding gene expression markers, PILOT uses limma's empirical Bayes approach (Ritchie et al, 2015) to contrast the expression of cells for a given cell type/clusters for two distinct groups of samples. In all cases, we use the Benjamini/Hochberg procedure to correct for multiple testing. Gene enrichment analysis is performed with g:profiler https://biit.cs.ut.ee/gprofiler/gost and includes correction for multiple testing based on the SCS algorithm (Reimand et al, 2007).

PILOT framework also provides statistical tests to evaluate the potential presence of batch artefacts in the data. In short, for clustering analysis, we use the Chi-Squared statistics to compare results with discrete variables, while for numerical variables this is based on ANOVA. For trajectory analysis, we compare discrete variables with disease progression scores with the ANOVA test and numerical variables with disease progression score with the Spearman correlation. We provide tutorials based on the kidney single-cell RNA-seq data on how this can be used to detect potential data artefacts.

## Data sets

We use public single-cell and pathomics data sets to benchmark the proposed methods (see Table 1).

### Single-cell data sets

Peng and colleagues (Peng et al, 2019) performed a single-cell study on pancreatic ductal adenocarcinoma (PDAC). They characterized 11 healthy and 24 PDAC samples with a total of 57,530 scRNA-seq, which were clustered and annotated in 10 major pancreas cell clusters. As only raw data was provided (obtained from GSA: CRA001160 https://ngdc.cncb.ac.cn/gsa/browse/CRA001160), we reanalyzed the data using Seurat and re-annotated cell clusters using the same marker genes (Appendix Fig. S11).

Systemic lupus erythematosus is a common type of lupus that, in the immune system, pounds its tissues, yielding overall rash and tissue injury in the acted organs (Perez et al, 2022). This single-cell

study characterized peripheral blood mononuclear cells (PBMCs) gene expression of 261 donors consisting of 1,263,676 cells. Of these donors, 99 were healthy controls, and 162 were disease patients. We use the normalized and clustered (11 groups) data sets provided in Gene Expression Omibus (GEO; GSE174188; GSE174188_CLUES1_adjusted.h5ad.gz).

We use single-cell and single-nucleus assays from kidney samples from the Kidney Precision Medicine Project (Lake et al, 2023). For this study, we used the single-cell RNA-seq experiments, including 76,020 cells, annotated in 57 major cell clusters. This data has 36 donors, including 18 control, 5 acute kidney failure, and 13 chronic kidney samples). For this data, we consider those acute kidney failure and chronic kidney samples infected with diabetes. Pre-processed data was obtained from GEO (GSE169285).

The lung cancer single-cell atlas is another large data set with 941,504 cells from 165 donors (Sikkema et al, 2023). The samples are formed by 51 normal lung, 18 chronic obstructive pulmonary diseases, 76 lung adenocarcinomas, 13 non-small cell lung carcinomas, and 12 squamous cell lung carcinoma. The cells are clustered in 33 clusters/cell types. The atlas is based on distinct studies measured under distinct platforms. We only consider here samples from lung tissue and measured with the 10X genomics platform. Data was obtained from Zenodo (ID 7227571).

We have recently proposed a large study to characterize myocardial infarction (Kuppe et al, 2020). We recovered 115,517 high-quality cells, which were clustered in 33 cellular clusters. We focus here on samples associated with healthy heart tissues (healthy controls and remote zones; $n = 13$) and ischemic zone (IZ; $n = 7$). We exclude one sample due to low quality (less than 1000 genes (on average) per cell). A final data set is provided in Zenodo (ID 7435911).

The single-cell pancreas diabetes data (Hrovatin et al, 2022) covers 264,235 mouse pancreatic islet cells (with 13 cell clusters) from 52 samples. This includes 12 endocrine pancreas disorders, 6 type-1 diabetes mellitus, 12 type-2 diabetes mellitus, and 22 normal samples. We excluded four samples (Embryos E12-E15) due to being outliers by trajectories of methods. The data set was obtained in GEO (GSE211799). Follicular lymphoma data (Han et al, 2022) collected by Han and colleagues containing 137,147 single cells have been processed to illustrate the various tumor and immune cell populations of Follicular lymphoma. Cells have been clustered into 18 cell clusters from 23 donors (3 normal and 20 lymphoma samples). The raw data is accessible from EGA (EGAS00001006052).

Finally, we obtained a study with PBMC cells from patients infected with Covid-19 (Ren et al, 2021). We only considered patients with PMBCs (frozen or fresh cells). This totals to 151 samples classified as either severe infection ($n = 70$), mild infection ($n = 61$), and control ($n = 20$). For these samples, we have 993,171 cells, which were grouped into 10 major cell clusters. The data was obtained from GEO (GSE158055).

### Pathomics data sets

The VALIGA study is a European cohort with kidney biopsies and accompanying clinical data of patients with IgA nephropathy (Coppo et al, 2014). We used a pathomics pipeline developed by us (Hölscher et al, 2023) to detect and measure 3 morphometric features of 65,483 tubules and 14 features associated with 24,227 glomeruli structures, which could be detected in 634 biopsies. Patients were classified regarding their glomerular filtration rate (GFR): normal (GFR > 60; $n = 400$), reduced ($30 < $ GFR $\leq 60$; $n = 177$) or low (GFR $\leq 30$; $n = 57$).

Lower GFR indicates lower kidney function. A second pathomics data set is the Aachen Cohort data, which includes 57 samples with healthy controls and distinct diseases/co-morbidity associated with lower kidney function (Hölscher et al, 2023). We analyzed the histology slides as before (Hölscher et al, 2023), which provided 4731 glomeruli and 46,999 tubules, each quantified with 3/14 morphometric features. Patients were classified as either being healthy controls ($n = 17$) or diseased ($n = 40$).

We employed a uniform pre-processing pipeline utilizing Seurat (Hao et al, 2021) for the normalization and clustering of structures. First, we normalized the data with the function NormalizeData of Seurat, and next ran the ScaleData function. Next, we performed dimension reduction (RunPCA) and kept the 10 and 2 main components for glomeruli and tubules, respectively. Next, clusters were found with the Leiden algorithm by using the FindNeighbors and FindClusters functions, respectively. The same pipeline was performed on the kidney IgAN (VALIGA) and Aachen cohort data. As morphometric features are not comparable between glomeruli and tubules, these data are analyzed independently, which results in four distinct data sets. At the patient level, PILOT also allowed the combination of the Wasserstein distance for glomeruli and tubuli, which provided a unique trajectory for each cohort.

Human pathology data collection and analysis in this study was performed in accordance with the Declaration of Helsinki and was approved by the local ethics committee of the RWTH Aachen University (EK-No. 315/19). All analyses were performed retrospectively in an anonymous fashion and the need for informed consent was waived by the local ethics and privacy committee for all datasets.

## Competing methods

### Pseudo-bulk

We investigate pseudo-bulk here as a baseline method for sample-level analysis. For single-cell data, we sum the gene expression count for all cells in a sample. Then the Poisson distance (Witten, 2011) between summed counts is employed to compute the sample contrasts. For pathomics data, we calculate the average morphological features per sample and then scale values between 0 and 1. Afterwards, the principal components(PCA) for samples are computed. We detected knees in the variance plots and kept the PCs with the highest variances. We calculated the cosine distance, which was used as input for diffusion maps and Leiden clustering.

### Proportions

Here, we take the original single cell and pathomics data and calculate the proportion of cell types (clusters) per sample. Next, we calculate the dissimilarity among samples based on their fraction of clusters by cosine distance. Finally, we apply the diffusion map to the distance matrix and get the order of samples.

### PhEMD

PhEMD required the use of Monocle 2 (Trapnell et al, 2014) to perform normalization, dimension reduction, and clustering of cells. The inputs of PhEMD for all data sets are the first 50 principal components of PCA and original pathomic measurements. Among other parameters, PhEMD/Monocle2 requires the definition of a distribution function for expression values. We use as default Negative Binomial distributions to model single-cell data sets as

advised in the tutorials. For PBMC COVID-19, Diabetic, and Lupus data, we used Gaussianff, due to the fact no cluster was found with the default model. For pathomics data, we used truncated normal distributions due to their best performance. We then used the distance matrices provided by PhEMD as input for a diffusion map analysis as implemented in PILOT.

### Evaluation of methods

We evaluate these methods in distinct scenarios: clustering, distance, and trajectory estimation. In the clustering evaluation problem, we evaluate the methods regarding their performance in clustering the data using graph-based clustering (Traag et al, 2019). The number of clusters were the same as the number of true classes in the data. The accuracy of the methods are evaluated with the well-known external clustering index, the adjusted Rand index (ARI) (Rand, 1971). This index has a value from $-1$ to 1, where values of 1 indicate a perfect match between the clustering and the labels, while values of 0 or lower indicate solutions found by chance. Next, we use the Silhouette index (Rousseeuw, 1987) to evaluate how well samples with the same labels are separated according to the distance estimated by each method (distance evaluation). This evaluation does not require a clustering analysis. The Silhouette index has values between 0 and 1, while 1 indicates a clear distance separation between the provided class labels. Finally, we evaluated all methods in relation to their prediction of disease trajectories, i.e., the relation of sampling ordering with the class labels. We use the Area Under the Precision-Recall Curve statistic (AUCPR; Trajectory Evaluation) for this. We only consider the two class problem (disease vs. non-disease). For each of these three scenarios, we use the Friedman–Nemenyi test to contrast the performance of the methods in distinct data sets. It allows us to compare the performance of several algorithms when they are evaluated on the same data sets. Here, the null hypothesis is that all algorithms have the same performance. The test is non-parametric and is based on the rank of the algorithm at each data set. Low rankings indicates best methods. This is important, as ARI values (or any other evaluation statistic) are data set specific, e.g., some clustering problems are more difficult than others. By evaluating the rank, the test indicates which methods perform relatively better than others. More precisely, the Friedman test is used to verify whether there is a significant difference in any of the ranks of any of the compared algorithms. It is then followed by the pair-wise Nemenyi test, which indicates the significance in difference between all pairs of algorithms. The last steps includes the correction for multiple testing (Demšar, 2006).

## Data availability

The datasets and computer code produced in this study are available in the following databases: Pre-processed R and H5ad objects used as input in benchmarking and case studies are deposited in zenodo, part 1 and zenodo, part 2. PILOT code, including documentation, tutorials, and scripts for replicating experiments, are found in https://github.com/CostaLab/PILOT and https://pilot.readthedocs.io.

# Peer review information

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

## Acknowledgements

We thank the European Validation Study of the Oxford Classification of IgAN (VALIGA), funded by the Immunonephrology Working Group of the European Renal Association for access to VALIGA data. This work was funded by grants of the Deutsche Forschungsgemeinschaft (DFG-GE 2811/3) to IGC; the clinical research unit KFO5011 (Projekt ID: 445703531) for IGC, PB, RK, and CK and by the Bundesministerium für Bildung und Forschung (BMBF e:Med Consortia Fibromap) to IGC and RK. PB is supported by the German Research Foundation (DFG, Project IDs 322900939, 454024652 and 432698239), European Research Council (ERC Consolidator Grant No. 101001791), and the Federal Ministry of Education and Research (BMBF, STOP-FSGS-01GM2202C).

## Author contributions

**Mehdi Joodaki**: Conceptualization; Resources; Data curation; Software; Formal analysis; Writing—original draft. **Mina Shaigan**: Conceptualization; Resources; Data curation; Software; Formal analysis; Writing—original draft. **Victor Parra**: Data curation; Software. **Roman D Bülow**: Conceptualization; Data curation; Formal analysis. **Christoph Kuppe**: Formal analysis; Writing—review and editing. **David L Hölscher**: Data curation; Formal analysis. **Mingbo Cheng**: Software. **James S Nagai**: Software. **Michaël Goedertier**: Software; Writing—review and editing. **Nassim Bouteldja**: Data curation. **Vladimir Tesar**: Data curation. **Jonathan Barratt**: Data curation. **Ian SD Roberts**: Data curation. **Rosanna Coppo**: Data curation. **Rafael Kramann**: Data curation; Funding acquisition; Writing—review and editing. **Peter Boor**: Conceptualization; Funding acquisition; Methodology; Project administration; Writing—review and editing. **Ivan G Costa**: Conceptualization; Funding acquisition; Investigation; Methodology; Writing—original draft; Project administration.

## Funding

## Disclosure and competing interests statement

The authors declare no competing interests.

# Expanded View Figures

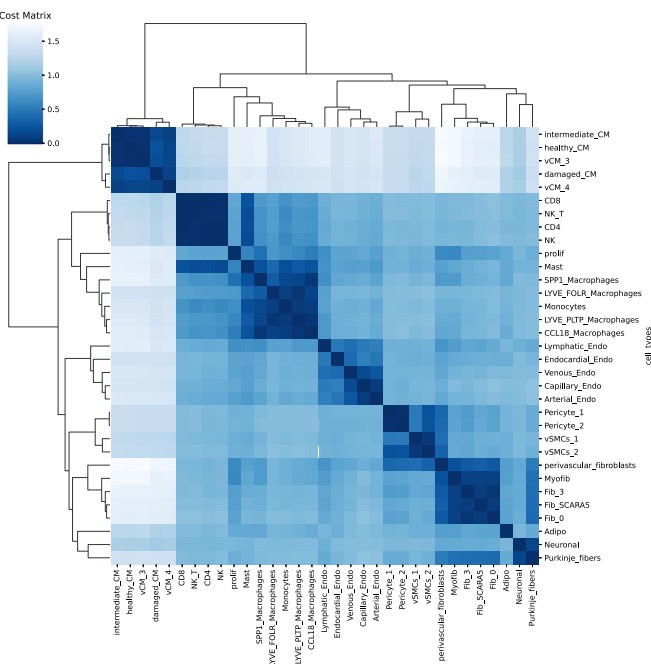

**Figure EV1.  Example of cost matrix used for optimal transport in the myocardial infarction scRNA-seq data.**

The cost matrix obtained at the analysis of the myocardial infarction based on 33 cell clusters. The cost (or distance) between cardiomyocyte cell types is lower to each other (healthy-CM, intermediary-CM, damaged-CM) than when compared with fibroblast cells.

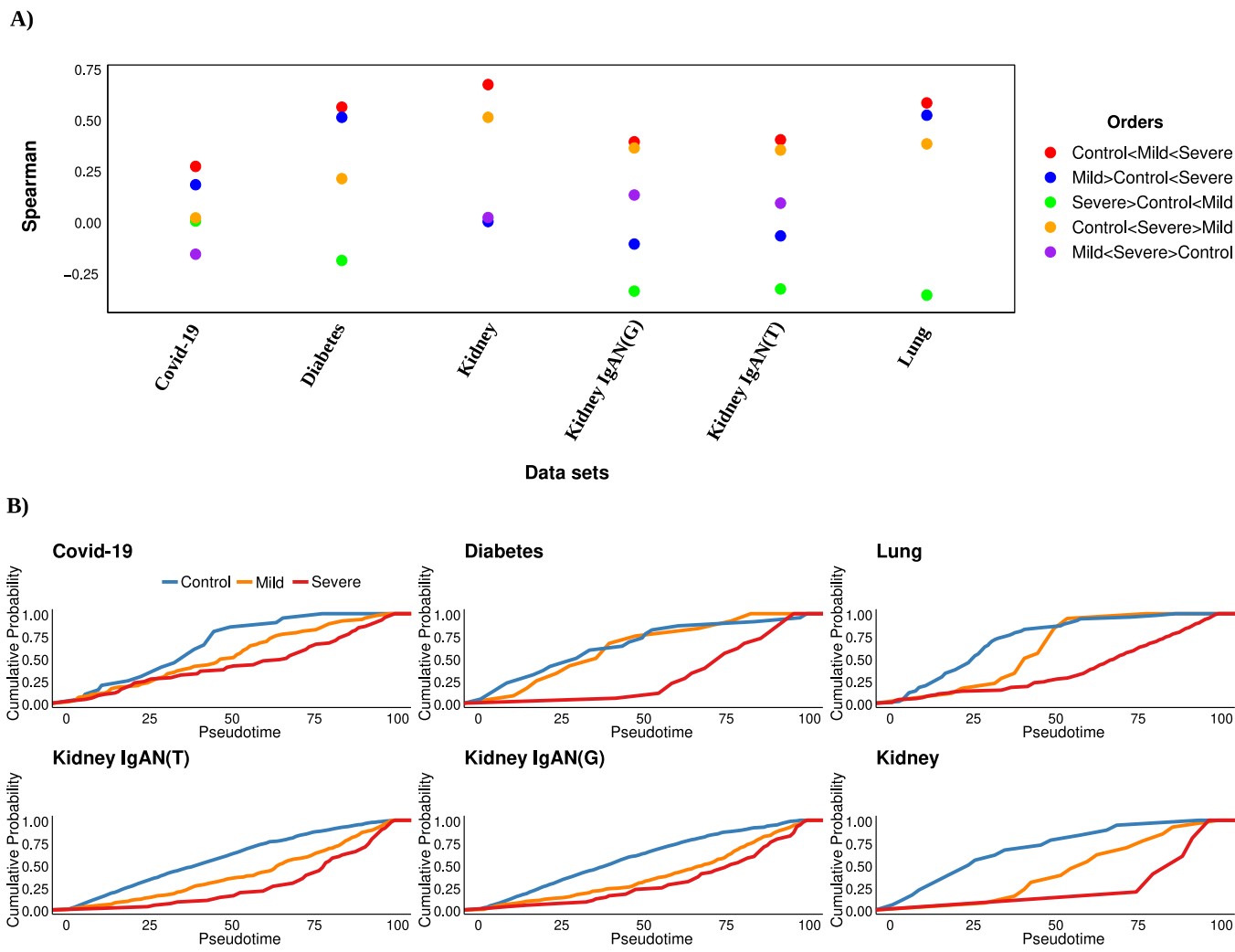

**Figure EV2. Benchmarking analysis of trajectories and disease progression stage.**

(A) Spearman Correlation (y-axis) between disease progression scores and ordered classes for distinct data sets (x-axis) by using PILOT. We systematically shuffled the order of control, mild and severe samples to investigate if the order is capture by the algorithms. We observe highest Spearman correlation values for the order "control < mild < severe" in all data sets. (B) Cumulative probability of control, mild and severe cases (y-axis) over PILOT estimated pseudotime (x-axis) for all multi-class data sets.

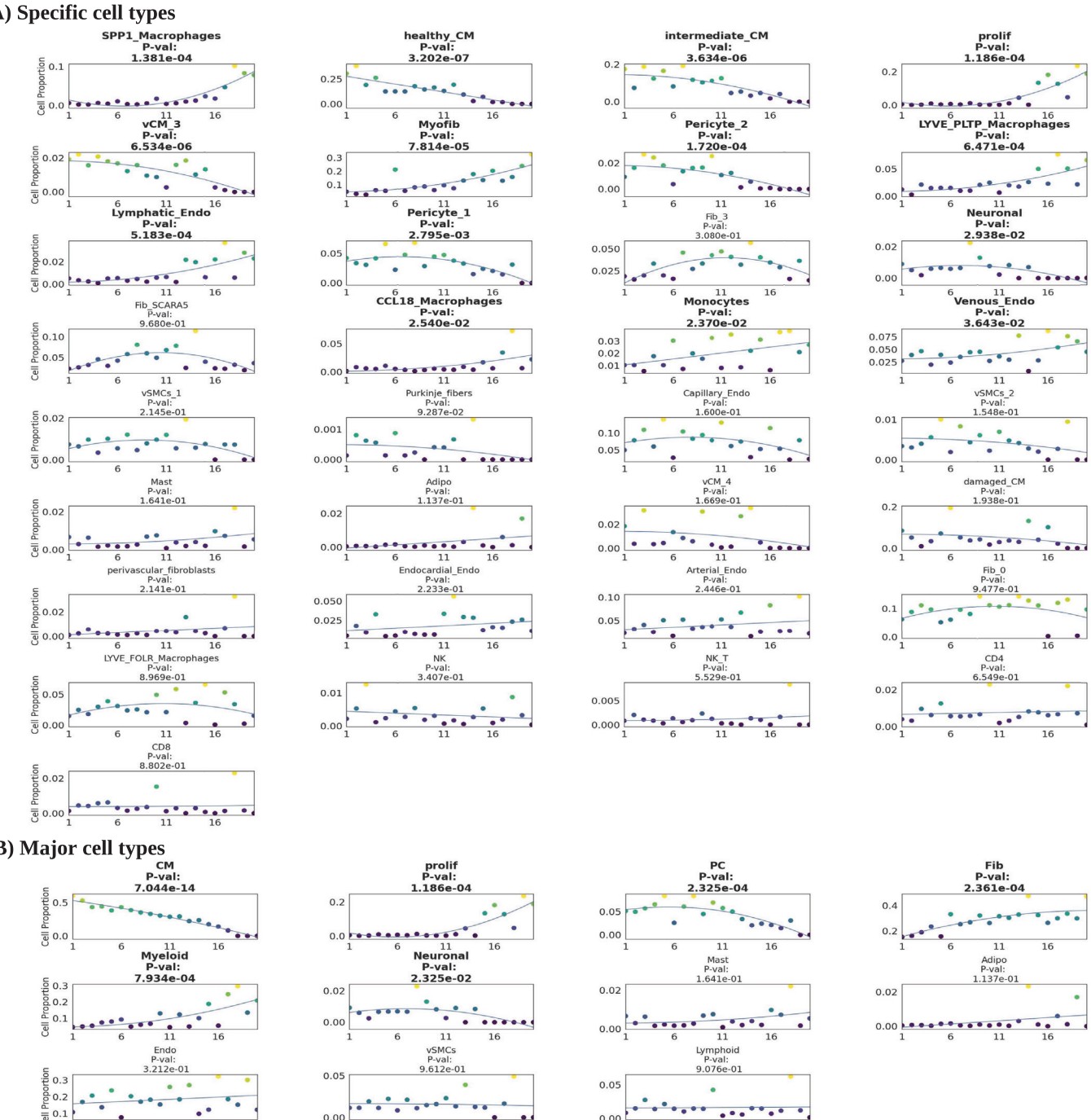

**Figure EV3. Cell composition changes in the myocardial infarction scRNA-seq data.**

Cell cluster frequency (y-axis) vs. PILOT disease progression (x-axis) for Myocardial Infarction scRNA-seq data for high granularity clusters (k = 33) (**A**) and low granularity clusters (k = 11) (**B**). Cells with a significant association with pseudotime are marked in bold. *p*-value of (**A**, **B**) were estimated with the F-statistic test.

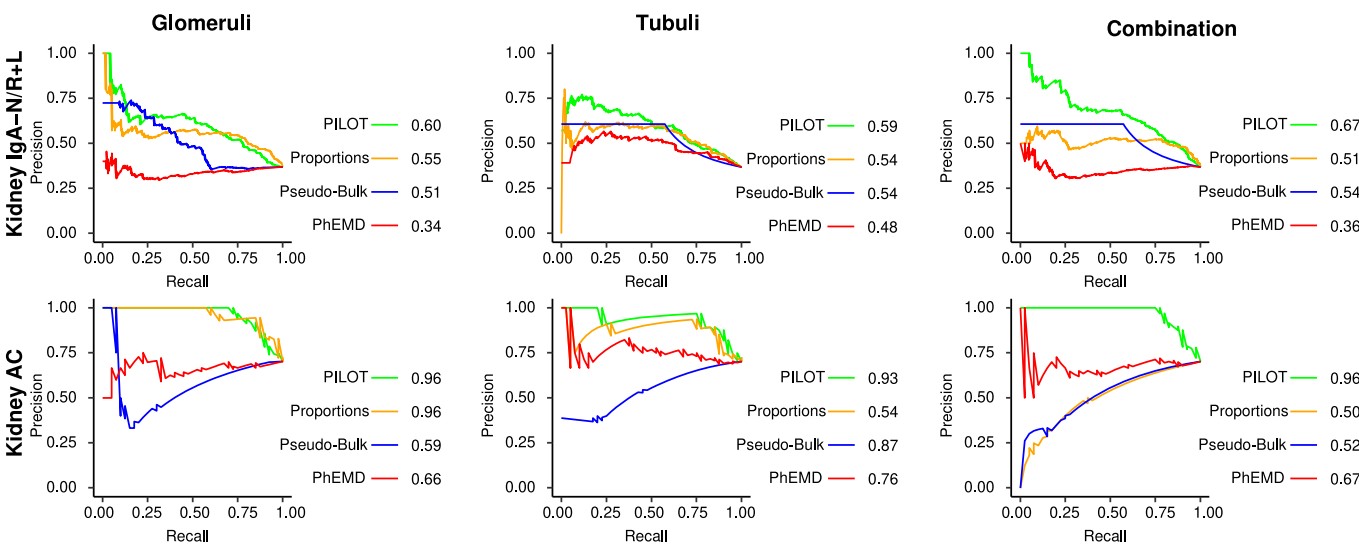

**Figure EV4. Trajectory prediction scores for Kidney pathomics data sets.**

AUCPR plots for Glomeruli, Tubule and both (combined) for Kidney AC, Kidney IgAN pathomics data.

