## [Peer Review File · Molecular Systems Biology]

Detection of Patient-Level distances from single cell genomics and pathomics with Optimal Transport

Mehdi Joodaki, Mina Shaigan, Victor Parra, Roman D. Buelow, Christoph Kuppe, David L. Holscher, Mingbo Cheng, James S. Nagai, Michaël Goedertier, Nassim Bouteldja, Vladimir Tesar, Jonathan Barratt, Ian Roberts, Rosanna Coppo, Rafael Kramann, Peter Boor, and Ivan Costa

DOI: 10.15252/msb.202311962

Corresponding author(s): Ivan Costa (ivan.costa@rwth-aachen.de), Peter Boor (pboor@ukaachen.de)

Review Timeline:

Transfer from Review Commons:	24th Aug 23
Editorial Decision:	26th Sep 23
Revision Received:	6th Nov 23
Editorial Decision:	14th Nov 23
Revision Received:	20th Nov 23
Accepted:	24th Nov 23

The logo for Review Commons, with "Review" in a large, blue, serif font and "COMMONS" in a smaller, blue, sans-serif font below it.

Editor: Poonam Bheda

Transaction Report: This manuscript was transferred to Molecular Systems Biology following peer review at Review Commons.

Review #1

1. Evidence, reproducibility and clarity:

Evidence, reproducibility and clarity (Required)

The paper describes a computational method, PILOT, that uses optimal transport to compute the Wasserstein distance between two individual single-cell samples. It uses PILOT to detect sample (patient) level trajectories and clusters associated with diseases. The method was applied separately to single-cell genomics data and to digital pathology data. The method was applied to several datasets and compared against other tools.

****Major comments:****

The paper is not easy to follow and should be improved considerably to make it readable and reproducible. Consequently, I was not convinced that the PILOT method is much better than other methods.

At first read of the title and abstract, I got the impression that the method analyzes single cell and pathomics data concurrently rather than separately. This should be fixed.

The usage of Wasserstein distance to compute distance between single-cell samples is elegant and is the main strength of this study. Given that PILOT is the main achievement, it should be described more carefully and in a detailed manner.

For example, in the first Results paragraph, "The test indicates for features explaining the predicted pseudotime by fitting either linear or quadratic models" - I could not understand this sentence. Also, which test do the authors refer to? A few sentences down, there is a reference to a Wald test, is that it?

One of the key aspects of the Wasserstein distance is the cost metric. The determination of the cost metric should be detailed as part of the Results. Have the authors considered and estimated other ways to define the distance?

Figure 1 provides a schematic view of PILOT. However, there is no explanation of the notation, which makes it confusing rather than helpful. Also, what is the relationship between J and j , if any?

The motivation and usage of adjusted Rand index (ARI) and Friedman-Nemenyi tests

should be provided. Currently, they are not clear, including why those tests are suitable in the cases shown.

Fig. 2 the use of method colors should be constant across panels. The proportions method works at least as well as PILOT in 2B and 2C (silhouette and AUPR). Explain why PILOT is better. Likewise, Figure 2C,D and Figures S1 and S2 don't show a clear and consistent advantage for PILOT over other methods. Explain what advantage of PILOT do the fraction panels highlight in Fig. 2E and Fig. S3. Fig. 2C is not mentioned in the text.

I assume Kidney IgAN (text) and Kidney IgA (fig. 2) are the same.

Fig. 3B fix the p-value notation (what is $p=1.05E?$) and R2 (R square?). Note that both this problem also occurs in other figs. Fig. 3B shows the major cellular changes. Are these changes consistent with known ones? Explain and provide references. Are there cell types that were expected to show a change and did not? Same questions for Fig. 3C wrt genes. Is this an exploratory analysis highlighting interesting candidate genes? If so, it should be described as such. The point of Fig. S6 and its major findings should be mentioned in the text (or it can be removed).

Fig. 4B legend - eGFR not GFR. What do the high-low values of Fig. 2B refer to? Fig. S12 is out of order in supp file.

AUCPR - explain.

The github looks like work in progress with many internal comments (eg, add ,edit, etc). I could not find the tutorials.

****Minor comments:****

Introduction: "Alternatively, trajectory analysis can be performed to uncover disease progression allowing the characterization of early disease events." Citations should be added (some appear later in the text).

"Currently, there are no analytical methods to compare two single cell experiments from the same tissue from two distinct individuals." There have been several comparisons among data from patients, (e.g. Cain et al, 2023), so the authors should be more careful/accurate in their statements.

"Except for PhEMD, all related methods^{9, 11, 12} require labels of patients for their analysis and cannot be used in the unsupervised analysis " - this sentence comes immediately after describing ref 13, which can be used in unsupervised analysis and

accordingly is not cited in this sentence. The authors did well in describing ref 13 (a bioRxiv paper), and its description should come after this sentence.

"These can be clusters", clustered?

" acquire an injury cell states" remove an.

"As for scRNA-seq, there is no analytical method which is able to compare two or more histological slides based on morphometric properties of their structures." The sentence seems to refer to pathomics, not to sc data as suggested in "As for scRNA-seq"

"Thus PILOT represents the first approach to detect unknown patient trajectories and clusters" patient clusters were also observed by others (eg ref 13, Cain et al).

Equation 7 - $\text{Cosine}(M_i, M_i)$ should be $\text{Cosine}(M_i, M_j)$

In the beginning of the Results, PILOT is not referred to as a package but as a researcher ("PILOT explores").

2. Significance:

Significance (Required)

In general, the paper is a Methods paper. Hence, likely audience includes computational biologists interested in methodologies, not to biologists interested in the actual findings.

Although I am among the likely audience, I was not convinced by the merits of the method, potentially due to the way the paper was written.

I do not have sufficient expertise to check the math.

3. How much time do you estimate the authors will need to complete the suggested revisions:

Estimated time to Complete Revisions (Required)

(Decision Recommendation)

Between 1 and 3 months

Yes

Review #2

1. Evidence, reproducibility and clarity:

Evidence, reproducibility and clarity (Required)

Joodaki et al. propose PILOT, a computational method for analysing single-cell genomics and pathomics data. PILOT is a method that enable clustering, trajectory analysis, and feature detection at a patient level using scRNA-seq data. This is an important task and represent the growing application of scRNA-seq to understand diseases and other perturbations to other biological systems. In particular, PILOT enables unsupervised analysis which alleviate the need of patient labels required by many alternative methods. We have the following comments for the authors' consideration.

1. A key consideration in dealing with scRNA-seq data at a patient level is the batch effect in the data. Typically, each patient sample may be treated as a "batch" especially when they are processed separately to obtain a scRNA-seq dataset that are subsequently combined with scRNA-seq datasets from other patients to form a single dataset. Analysing these data without batch correction may lead to the identification of "cell types" and "states" that are driven by batch effect. In Figure 1. PILOT takes a clustered and integrated scRNA-seq data as input for analysis. I wonder how PILOT would behave if there is a strong batch effect in the data and how would the authors propose to handle them?
2. It appears that the Wasserstein distance (W) matrix of the samples was used for

patient clustering and also trajectory analysis. However, most of the figures presented in the manuscript are for trajectory analysis. Since the patient clustering were performed prior to trajectory analysis, could the authors visualize the patient data based on the W before performing disease trajectory estimation?

3. In trajectory analysis in figure 2D and E, why not use Multi-scale PHATE which appears to be specifically designed for trajectory analysis? The authors also mentioned SCANCell. While these methods require labels of patients for their analysis, it would be interesting to know how well they perform in comparison to PILOT if such information is available.

4. The current design of PILOT appears to assume that there is always a "smooth" trajectory in the data. Is this going to be the case in reality? What if we have a well separated and distinct groupings of the patients and controls data? In the latter case, imposing a trajectory seems artificial. I am also not sure how meaningful the trajectory analysis would be if, biologically, such a smooth transition is not present in the data.

5. The feature analysis is also built on trajectory analysis using regression models. Again, how would this work out if there isn't a smooth trajectory/transition in the data (e.g. the data are obtained from a discrete case-control study)?

6. It is not clear from the formulation of PILOT (and also Figure 1) if the cell type labels is required/used or the cluster id of a clustering algorithm was used instead. The author also mentioned that the clustering output does not have much impact on the downstream analysis. I wonder why and if so can we group the data in any way we want for downstream analysis? This can be useful when one would like to focus on certain grouping of cells.

2. Significance:

Significance (Required)

PILOT is designed for analysing scRNA-seq data at a patient level. There is a growing application of scRNA-seq to diseases and the development of computational tools for analysing such data at phenotype level is critical. The key aspect of PILOT compared to other currently available tools is that it enables unsupervised analysis which alleviate the need of patient labels required by many alternative methods.

3. How much time do you estimate the authors will need to complete the suggested revisions:

Estimated time to Complete Revisions (Required)

(Decision Recommendation)

Between 1 and 3 months

Yes

Full Revision

Manuscript number: RC-2023-02028

Corresponding author(s): Ivan G Costa

1. General Statements

We would like to express our gratitude to the referees for their meticulous review of the manuscript.

The utilization of single-cell genomics and pathomics, i.e., unbiased, large-scale quantification of morphological features in histopathology, generates large data within a single patient. The next stage in clinical research is the application of these methods on large patient cohorts. However, the multi-scale properties of this data (samples are represented as clusters of cells) make such analysis highly challenging. Here we describe PILOT - a novel computational framework addressing this challenge. It explores the optimal transport framework to measure distances between multi-scaled data. This allows unsupervised analysis such as clustering and trajectory inference at a sample (patient) level. Moreover, we propose the use of robust regression and statistical models to interpret predictions across scales, i.e., finding genes, cell types, or morphological features associated with clusters/disease trajectories. We are not aware of any other computational approaches tackling these problems in a single framework.

We have addressed all the queries and suggestions put forth by the reviewers. Notable enhancements made include:

1. **Expansion of PILOT Functionality and Analysis:** We have substantially extended the functionality and analysis capabilities of PILOT, particularly in relation to sample clustering. This enhancement now encompasses the incorporation of statistical tests aimed at identifying cell types and genes associated with distinct patient groups. We applied this expanded feature in an exploratory analysis of sub-clusters within pancreas ductal adenocarcinoma data (PDAC).
2. **Clarification of Benchmarking Methods:** We have provided clear elucidations of the methods employed for benchmarking PILOT alongside competing methodologies. Our benchmarking approach is notably comprehensive, encompassing twelve different datasets and evaluating four to five competing methods through statistical assessment across three problem domains: clustering, distance measurement, and trajectory estimation. The outcomes of these evaluations consistently demonstrate the superior performance of PILOT's Wasserstein metric across all three problem domains. It is noteworthy that previous studies have often limited their analyses to exploratory evaluations on individual datasets, lacking the level of comprehensive benchmarking undertaken in this study.
3. **Examination of Various Factors:** We have conducted a thorough investigation into the impacts of batch correction, cluster/cell type resolution, and parameter choices used within the PILOT framework.

4. **Enhancement of Text Description:** We have enhanced the textual descriptions to provide a high-level overview of the PILOT methodology, along with justifications for the methodological decisions made.
5. **Improvement of Code and GitHub Repository:** To enhance accessibility and promote reproducibility, we have made improvements to the codebase and the associated GitHub repository.

In summary, PILOT stands as a distinctive and all-encompassing framework. It holds the unique distinction of being the sole method offering comprehensive tools for both clustering and trajectory analysis of samples within multiscale single-cell and pathomics data. Moreover, it incorporates statistical methodologies for the interpretation of results. The effectiveness of these tools has been thoroughly validated through the most extensive benchmarking study performed to date on sample-level analysis. The versatility of PILOT is demonstrated through its successful application in exploratory analyses of three distinct datasets: elucidating trajectories in myocardial infarction single-cell RNA-seq data, uncovering trajectories within pathomics data from kidney IgNA patients, and facilitating the clustering of pancreas adenocarcinoma samples. We firmly believe that these contributions hold significant value for the fields of bioinformatics, single-cell genomics, and pathology.

Reviewer #1 (Evidence, reproducibility and clarity (Required)):

The paper describes a computational method, PILOT, that uses optimal transport to compute the Wasserstein distance between two individual single-cell samples. It uses PILOT to detect sample (patient) level trajectories and clusters associated with diseases. The method was applied separately to single-cell genomics data and to digital pathology data. The method was applied to several datasets and compared against other tools.

Major comments:

The paper is not easy to follow and should be improved considerably to make it readable and reproducible. Consequently, I was not convinced that the PILOT method is much better than other methods.

We extend our appreciation to the reviewer for their valuable suggestion. We have further refined the manuscript by incorporating a comprehensive and high-level description of our method. This expansion encompasses methodological justifications and clarifications to enhance the overall clarity. Additionally, we wish to emphasize that, to the best of our knowledge, our benchmarking analysis stands as the most comprehensive within the current literature. The results of this analysis unequivocally demonstrate that PILOT surpasses all competing methods in at least one of the various computational analysis tasks, namely clustering, trajectory estimation, and distance evaluation.

Furthermore, we have undertaken significant enhancements in the codebase of PILOT, coupled with a reorganization of the associated GitHub repository. This effort includes the development of

in-depth and improved tutorials that faithfully replicate the analyses conducted on datasets related to myocardial infarction, pancreas adenocarcinoma, and pancreas pathomics (<https://pilot.readthedocs.io/en/latest/>). This changes guarantee the reproducibility of the PILOT framework.

See below for specific changes and additional clarifications.

At first read of the title and abstract, I got the impression that the method analyzes single cell and pathomics data concurrently rather than separately. This should be fixed.

We have changed the text of the abstract and introduction to make clear that PILOT is either applied to single cell or pathomics data independently.

The usage of Wasserstein distance to compute distance between single-cell samples is elegant and is the main strength of this study. Given that PILOT is the main achievement, it should be described more carefully and in a detailed manner.

For example, in the first Results paragraph, "The test indicates for features explaining the predicted pseudotime by fitting either linear or quadratic models" - I could not understand this sentence. Also, which test do the authors refer to? A few sentences down, there is a reference to a Wald test, is that it?

PILOT has three major parts: (1) a method for measuring distance of samples with optimal transport; (2) an patient level unsupervised analysis part (clustering or trajectory analysis) and (3) a part for explaining predicted trajectories/clustering. The sentence mentioned before, refers to the interpretation approach after trajectory analysis. Here, we fit linear, quadratic or linear quadratic models to find association of predicted sample pseudo-time with data features (gene expression values in scRNA or morphological features in pathomics data). This fit can be done for all cells in the data or only for cells from a specific type. In the case of a cell specific fit, we use a Wald test to check if the cell type fit differs from all other cell types in the data, i.e. the gene is associated with the trajectory and the expression changes are specific to the cluster at hand.

While these details were found in the method section, we agree with the referee that they can be better introduced in the main manuscript. We have therefore improved the first subsection of the results and Figure 1 to reflect this.

One of the key aspects of the Wasserstein distance is the cost metric. The determination of the cost metric should be detailed as part of the Results. Have the authors considered and estimated other ways to define the distance?

This is an interesting question. Currently, PILOT uses the Cosine metric. In our revision, we evaluate other metrics (Euclidean, Manhattan, and Chebyshev). This benchmarking indicates that

the Cosine and Manhattan performed best regarding the clustering problem (ARI), while Cosine was better than all metrics for the Silhouette statistic; and Cosine and Euclidean performed best regarding AUPR. Therefore, we adopt the Cosine metric in the paper. We include these results in the revised manuscript and in Sup. Fig. 5F-H.

Figure 1 provides a schematic view of PILOT. However, there is no explanation of the notation, which makes it confusing rather than helpful. Also, what is the relationship between J and j, if any?

We understand that the figure 1 was problematic, as it did not introduce the formulation. We have now improved the first sub-section of the results page and figure 1 to improve this.

The motivation and usage of adjusted Rand index (ARI) and Friedman-Nemenyi tests should be provided. Currently, they are not clear, including why those tests are suitable in the cases shown.

The adjusted Rand index is a well known metric to evaluate clustering results when labels are known. Among others this metric has many interesting features as it does not require an association of clusters with class labels. Moreover, it has a correction for random clustering solutions, therefore values lower than zero indicate poor solutions and values of 1 a perfect solution.

The Friedman-Nemenyi test allows us to compare the performance of several algorithms whenever evaluated in the same data sets. Here, the null hypothesis is that all algorithms have the same performance (same ARI statistic). The test is nonparametric and is based on the rank of the algorithm at each data set. This is important, as ARI values (or any other evaluation statistic) are data set specific, e.g. some clustering problems are more difficult than others. By evaluating the rank, the test indicates which methods perform relatively better than others. Moreover, it follows a rigorous statistical framework including correction for multiple testing. This test has an increasing adoption in the machine learning community (Demsar et al., JMLR, <https://jmlr.org/papers/v7/demsar06a.html>).

We have added phrases with these justifications in the main text (subsection Evaluation of patient-level clustering and trajectory analysis) and included a new section in the materials and methods with more information in the experimental design of the benchmarking analysis.

Fig. 2 the use of method colors should be constant across panels.

We have changed the colors of panels in figure 2A-C (and equivalent panels everywhere else) to avoid confusions.

The proportions method works at least as well as PILOT in 2B and 2C (silhouette and AUPR). Explain why PILOT is better.

The benchmarking analysis shows that PILOT has the highest ARI value (clustering performance) at absolute and ranking levels (Fig. 2A). Moreover the Friedman-Neymeni test indicates this PILOT has significantly higher ranking than all evaluated methods. Regarding Silhouette (distance evaluation) and AUPR (trajectory evaluation) both proportion and PILOT have similar absolute values (Fig. 2B and 2C; panel left), while PILOT has a superior ranking in both cases (Fig. 2B and 2C panel right). Friedman-Neymeni test indicates higher ranking of PILOT than PhEMD for Silhouette and higher ranking of PILOT than PhEMD and pseudo-Bulk regarding trajectory evaluation. The difference in the results on absolute and ranking values can be understood by looking at the statistics in table Table S1. PILOT has highest AUPR in 8 out 12 data sets; proportion has highest values in 5 (including 4 ties with PILOT); proportion-PHATE had 3 best results (including 3 ties with both PILOT and proportions), while PhEMD is best in one data set and Pseudo-bulk in 3 (including 1 tie with PILOT). Altogether, PILOT obtained a higher or equal AUCPR in 9 out of the 12 data sets. We have also changed Fig.2A, 2B and 2C to include all data points and to show the mean, as this provides a better visualization of the previously reported results.

Altogether, these results indicate that PILOT outperforms all competing methods in at least one of the evaluated problems (clustering, trajectory and distance estimation) and ranks favorably in all evaluated scenarios. We have changed the manuscript text to reflect these results.

Likewise, Figure 2C,D and Figures S1 and S2 don't show a clear and consistent advantage for PILOT over other methods. Explain what advantage of PILOT do the fraction panels highlight in Fig. 2E and Fig. S3. Fig. 2C is not mentioned in the text.

Figure 2D, 2E, and now figures S2 and S3 represent visualizations of the results, which were statistically evaluated in panels of Fig.2A-2C. As discussed in the previous point, PILOT does perform better than all methods for the clustering problem and performs better or as good as the proportion test on 9 of the 12 evaluated data sets in the trajectory problem. We also have improved the text to include references to all figures in the main text.

I assume Kidney IgAN (text) and Kidney IgA (fig. 2) are the same.

The correct name is IgAN and this has been corrected in Figure 2.

Fig. 3B fix the p-value notation (what is $p=1.05E?$) and R2 (R square?). Nrte tha both this problem also occurs in other figs. Fig. 3B shows the major cellular changes.

We now adopt the term "R-squared" in the figures. Also, the previous version did not display p-values properly. We apologize for this. This has been fixed now.

Are these changes consistent with known ones? Explain and provide references. Are there cell types that were expected to show a change and did not? Same questions for Fig. 3C

wrt genes. Is this an exploratory analysis highlighting interesting candidate genes? If so, it should be described as such.

Cardiac remodeling after myocardial infarction is characterized by loss of cardiomyocytes, infiltration by immune cells (myeloid and lymphocytes) and increase in myofibroblast populations (doi.org/10.1038/s41392-022-00925-z;doi.org/10.3389/fcvm.2019.00026). PILOT indicates these populations, with the exception of lymphocytes, are most relevant at both clustering levels (see Sup. Fig, 6). Particularly important are results from the low granularity analysis, as this indicates particular macrophage/fibroblast sub-populations (SPP1+ Mac. and Myofibroblast) with increase in disease. PILOT could not detect changes in lymphocyte cells, but this is explained by the poor coverage of these cells in the data set (>3%). We have updated the main manuscript to reflect this.

We also explicitly mention that the analysis of genes and cells are exploratory analysis.

The point of Fig. S6 and its major findings should be mentioned in the text (or it can be removed).

We now make the reference to the gene ontology analysis presented in the **new** Figure S7 more explicit in the text.

Fig. 4B legend - eGFR not GFR. What do the high-low values of Fig. 2B refer to?

We have fixed these points. High and low values of panel 4B refer to the eGFR.

Fig. S12 is out of order in supp file.

This has been fixed.

AUCPR - explain.

The AUCPR stands for area under the curve of the precision recall (AUCPR) curve. We have now improved the explanation of the evaluation metric in the main text and methods section.

The github looks like work in progress with many internal comments (eg, add ,edit, etc). I could not find the tutorials.

We have removed all the comments, improved the repository organization and code. The tutorials are explicitly mentioned in the main github page (<https://github.com/CostaLab/PILOT/>) and in readthedocs webpage (<https://pilot.readthedocs.io/>). It include tutorials replicating analysis with trajectory inference and clustering problems, which are discussed in the manuscript.

In the process of code review, we have noticed that while we could replicate all the analysis, the procedure for selection of healthy cardiomyocyte genes was distinct (gene were ranked by regression model fit p-value) than the analysis of the myofibroblast genes (genes were ranked by the Wald test p-value). As explained before, the Wald test, which compares the expression of the regression model fits across samples, is a more appropriate criteria, as it finds cluster and trajectory specific genes. We have changed the analysis of the cardiomyocyte to make the gene selection to be based on the Wald-test p-value. New results recover other sarcomere related genes (MYBPC3 and MYOM1) as being dysregulated during disease progression. These findings are in accordance with observations made in the original study presenting the data (Kuppe et al. 2022). We have updated Fig.3 and respective genes accordingly.

Minor comments:

Introduction: "Alternatively, trajectory analysis can be performed to uncover disease progression allowing the characterization of early disease events." Citations should be added (some appear later in the text).

We included a reference to PhEMD.

"Currently, there are no analytical methods to compare two single cell experiments from the same tissue from two distinct individuals." There have been several comparisons among data from patients, (e.g. Cain et al, 2023), so the authors should be more careful/accurate in their statements.

We assume that the referee mentions <https://www.nature.com/articles/s41593-023-01356-x>. Indeed, we were not aware of this recently published study. The manuscript focuses on comparing cell proportion changes (estimated by deconvolution) between distinct phenotypes and does not provide any approach for sample level analysis of single cell data. This is in our view a different problem than the one addressed by PILOT or PhEMD. We refer to it in our manuscript, as its cell community based analysis is an interesting approach for the interpretation of PILOT results.

"Except for PhEMD, all related methods^{9, 11, 12} require labels of patients for their analysis and cannot be used in the unsupervised analysis " - this sentence comes immediately after describing ref 13, which can be used in unsupervised analysis and accordingly is not cited in this sentence. The authors did well in describing ref 13 (a bioRxiv paper), and its description should come after this sentence.

We changed the text to reflect this.

"These can be clusters", clustered?

Done.

Full Revision

" acquire an injury cell states" remove an.

Done.

"As for scRNA-seq, there is no analytical method which is able to compare two or more histological slides based on morphometric properties of their structures." The sentence seems to refer to pathomics, not to sc data as suggested in "As for scRNA-seq"

This has been rephrased.

"Thus PILOT represents the first approach to detect unknown patient trajectories and clusters" patient clusters were also observed by others (eg ref 13, Cain et al).

This has been rephrased.

Equation 7 - Cosine(Mi,Mi) should be Cosine(Mi,Mj)

Done.

In the beginning of the Results, PILOT is not referred to as a package but as a researcher ("PILOT explores").

This has been rephrased.

Reviewer #1 (Significance (Required)):

In general, the paper is a Methods paper. Hence, likely audience includes computational biologists interested in methodologies, not to biologists interested in the actual findings.

Although I am among the likely audience, I was not convinced by the merits of the method, potentially due to the way the paper was written.

I do not have sufficient expertise to check the math.

In this revision, we have significantly enhanced the text to incorporate high-level descriptions of methods tailored for non-computational experts. Additionally, we have refined the description of the benchmarking process, which, as far as our knowledge extends, stands as the most comprehensive in the literature. This comprehensive analysis strongly underscores the statistical superiority of PILOT when compared to other methods. Lastly, PILOT presents an unique framework, encompassing methods for trajectory analysis, clustering, and interpretation of sample-level analyses within the realm of multiscale single-cell genomics and pathomics data.

Reviewer #2 (Evidence, reproducibility and clarity (Required)):

Joodaki et al. propose PILOT, a computational method for analysing single-cell genomics and pathomics data. PILOT is a method that enable clustering, trajectory analysis, and feature detection at a patient level using scRNA-seq data. This is an important task and represent the growing application of scRNA-seq to understand diseases and other perturbations to other biological systems. In particular, PILOT enables unsupervised analysis which alleviate the need of patient labels required by many alternative methods. We have the following comments for the authors' consideration.

1. A key consideration in dealing with scRNA-seq data at a patient level is the batch effect in the data. Typically, each patient sample may be treated as a "batch" especially when they are processed separately to obtain a scRNA-seq dataset that are subsequently combined with scRNA-seq datasets from other patients to form a single dataset. Analysing these data without batch correction may lead to the identification of "cell types" and "states" that are driven by batch effect. In Figure 1. PILOT takes a clustered and integrated scRNA-seq data as input for analysis. I wonder how PILOT would behave if there is a strong batch effect in the data and how would the authors propose to handle them?

This is an interesting question. Currently, PILOT is using the batch procedure used in the paper proposing the original data. We evaluate now the impact of batch correction methods implemented in scanpy (Harmony, bknn and Scanorama). We focus here on single cell data, which we have access to the original count matrix (Lupus, COVID, and Diabetes). We observe no impact of the batch correction algorithm in these data sets (see Sup. Fig. 5C-E). These results are now included in the manuscript.

We have noticed however that strong batch effects in the lung cell atlas or the kidney cell atlas. For the lung cell atlas, we observed that single cell data measured from distinct techniques (Seq-well, Drop-seq, 10x 5' and 10x 3') or distinct tissue sampling approaches confounded results for all evaluated approaches. Therefore, we restricted the analysis to the technology with more samples (10x genomics 3') and to lung tissues. This sample selection was previously described in the material and methods. Of note, the use of samples from distinct 10x genomic version kits (v1, v2 or v3) did not impact results. For the kidney cell atlas, we also observed a strong batch between single nuclei and single cell protocols. Here, we opted to focus on the largest cohort of single cell RNA experiments (see Review Fig. 1). Altogether, PILOT and other evaluated methods do require samples to be analyzed with an uniform technique and sampling approaches. We now include a brief discussion about this open point in the "Discussion" section. This is an important topic of future research.

Review Fig. 1. - Data of the Kidney Precision Medicine Project was measured using either single cell or single nucleus protocols. All evaluated methods were affected by the differences in these technologies and could not separate disease status in this data.

2. It appears that the Wasserstein distance (W) matrix of the samples was used for patient clustering and also trajectory analysis. However, most of the figures presented in the manuscript are for trajectory analysis. Since the patient clustering were performed prior to trajectory analysis, could the authors visualize the patient data based on the W before performing disease trajectory estimation?

Indeed, despite the clustering-based analysis (ARI statistics; Fig. 2A) the current manuscript focuses on results of the trajectory analysis. We now include additional features for clustering analysis. This includes heatmap visualizations of the OT distance matrices together with Leiden clustering (Sup. Fig. 1). See points 4 and 5 below for further changes regarding clustering analysis.

3. In trajectory analysis in figure 2D and E, why not use Multi-scale PHATE which appears to be specifically designed for trajectory analysis? The authors also mentioned SCANCell. While these methods require labels of patients for their analysis, it would be interesting to know how well they perform in comparison to PILOT if such information is available.

This is an interesting point. Multiscale-PHATE is based on doing a multi-resolution clustering of the cells. It then applies PHATE (instead of diffusion map) to find a non-linear embedding on the cell proportions across samples and resolutions. While this analysis is presented at Multiscale-PHATE manuscript (Fig. 5), we could not find any code or functionality in their github to replicate this (https://github.com/KrishnaswamyLab/Multiscale_PHATE). Moreover, we were not able to find a function to find the cluster/resolution associations of cells to reimplement the above mentioned analysis following the descriptions of the manuscript. We also contacted authors, but obtained no reply. It is also important to state that Multi-PHATE used a supervised filter to select cell types for further analysis.

Alternatively, we now include an evaluation of the use of cell proportions followed by a PHATE embedding in the trajectory based evaluation, which is close to the method proposed in Multiscale-PHATE. Our benchmarking indicates that Multiscale-PHATE is the third best ranked method being overpassed by proportion and PILOT. Regarding SCANCell, it focuses on the interpretation of cell communities and it uses embedding/distances by exploring PhEMD. Therefore, its performance in the trajectory or clustering performance problem is the same as PhEMD. We refer to these points in the text now.

4. The current design of PILOT appears to assume that there is always a "smooth" trajectory in the data. Is this going to be the case in reality? What if we have a well separated and distinct groupings of the patients and controls data? In the latter case, imposing a trajectory seems artificial. I am also not sure how meaningful the trajectory analysis would be if, biologically, such a smooth transition is not present in the data.

The EMD based distance can be used both for clustering or trajectory analysis. Also, PILOT performed quite well in the clustering problem benchmarking (Fig. 2A). The choice of application lies on the problem at hand. In our view, both the kidney pathomics and the myocardial infarction data (explored in Fig. 3 and Fig. 4) represent medical data with potential disease trajectories. We now expand the PILOT framework to include new visualizations and statistical methods to improve the interpretation of the clusterings (see point 2; Fig. S1; and point 5 and new Fig. 5).

5. The feature analysis is also built on trajectory analysis using regression models. Again, how would this work out if there isn't a smooth trajectory/transition in the data (e.g. the data are obtained from a discrete case-control study)?

We expanded the PILOT framework to also include statistical tests for accessing changes in cell populations and markers for the clustering problem. First, we use a Welch's t-test to evaluate cell proportion changes associated with detected clusters. Next, we use a differential analysis test from limma to find genes within a cluster, whose gene expression is changing between the two groups of samples for a given cluster of cells. While these are standard approaches in the literature, this further improves the functionality of PILOT as a general framework for patient level analysis. This is now described in PILOT manuscript (Results subsection "Patient level distance with Optimal Transport" and methods).

We also include in the manuscript an explorative analysis of a sub-cluster found in the PDAC data. This analysis could find a population of PDAC patients displaying higher levels of malignant cells and marked by both increase in hypoxia and fibrosis pathways. This example highlights how PILOT can be used to find potentially interesting groups of samples. These are implemented in the main manuscript (Fig. 5). We also include a new tutorial of PILOT with this analysis (see <https://pilot.readthedocs.io/>).

6. It is not clear from the formulation of PILOT (and also Figure 1) if the cell type labels is required/used or the cluster id of a clustering algorithm was used instead. The author also mentioned that the clustering output does not have much impact on the downstream analysis. I wonder why and if so can we group the data in any way we want for downstream analysis? This can be useful when one would like to focus on certain grouping of cells.

Clustering at the cells (or structure level) is required. For the benchmarking analysis, we have used the cell annotation reported in the original paper, which were derived via clustering analysis. The use of annotated clusters is crucial for interpretation. We also included in the original manuscript an analysis on the impact of the clustering resolution of the Leiden algorithm. This indicates that the change in resolution did not have a high impact in the clustering (ARI) of the samples (Sup. Fig. 5A-B). However, this analysis could not consider any interpretation of results, as cluster labels were not present.

We believe, however, that the granularity of the clustering will impact the interpretation of the sample analysis. To investigate this, we evaluate how using higher level annotation/clustering of the heart myocardial infarction (also reported in Kuppe et al. 2022) impacts our ability to find cell specific changes. We observe similar changes whenever using low resolution clustering (decrease of cardiomyocytes, increase in fibroblasts and myeloid cells). However, this analysis loses a lot of important nuances found in the high resolution clustering (see Sup. Fig. 6). For example, it does not recover the fact that damaged cardiomyocyte populations have a slower decay than healthy myocytes. Or the fact that myofibroblasts has an increase in the latter disease trajectory stage, while progenitor fibroblast cells (Fibro_Scara5) have an increase previous to myofibroblasts. These results show how low resolution clustering can lead to loss of interesting information contained in cellular sub-states or cell sub-populations. This is now discussed in the results subsection 'PILOT trajectories detect events associated with cardiac remodeling in myocardial infarction'.

Reviewer #2 (Significance (Required)):

PILOT is designed for analyzing scRNA-seq data at a patient level. There is a growing application of scRNA-seq to diseases and the development of computational tools for analyzing such data at phenotype level is critical. The key aspect of PILOT compared to other currently available tools is that it enables unsupervised analysis which alleviate the need of patient labels required by many alternative methods.

Full Revision

Thanks for this very positive feedback and constructive comments.

26th Sep 2023

Manuscript Number: MSB-2023-11962

Title: Detection of Patient-Level distances from single cell genomics and pathomics data with Optimal Transport (PILOT)

Dear Prof. Costa,

Thank you for submitting your manuscript to Molecular Systems Biology. My colleague Poonam Bheda, handling editor of your manuscript, is out of office and in the interest of time I took over handling your manuscript. We have now heard back from the two reviewers who were asked to evaluate your revision. These are the same reviewers who evaluated the initial version of the manuscript at Review Commons. As you will see below, the reviewers think that the study has improved as a result of the performed revisions. They do however list some remaining issues, which we would ask you to address in a revision. We would also ask you to address some editorial issues listed below.

- Please provide a .doc version of the manuscript text (including legends for main figures and EV figures) and individual production quality figure files for the main and EV Figures (one file per figure).

- We have replaced Supplementary Information by the Expanded View (EV format). In this case, all additional figures can be included in a PDF called Appendix. Appendix Figures should be labeled and called out as: "Appendix Figure S1, Appendix Figure S2, ... etc.". Each Appendix Figure legend should be provided below the corresponding Figure in the Appendix. Please include a Table of Contents in the beginning of the Appendix. For detailed instructions regarding expanded view please refer to our Author Guidelines: .

All 9 additional Figures should be included in the Appendix.

- Please provide a "standfirst text" summarizing the study in one or two sentences (approximately 250 characters), three to four "bullet points" highlighting the main findings and a "synopsis image" (550px width and max 400px height, jpeg format) to highlight the paper on our homepage.

- All Materials and Methods need to be described in the main text. We would encourage you to use 'Structured Methods', our new Materials and Methods format. According to this format, the Material and Methods section should include a Reagents and Tools Table (listing key reagents, experimental models, software and relevant equipment and including their sources and relevant identifiers) followed by a Methods and Protocols section in which we encourage the authors to describe their methods using a step-by-step protocol format with bullet points, to facilitate the adoption of the methodologies across labs. More information on how to adhere to this format as well as downloadable templates (.doc or .xls) for the Reagents and Tools Table can be found in our author guidelines: . An example of a Method paper with Structured Methods can be found here:

- Please include a Data availability section, structured according to the example below:

The datasets and computer code produced in this study are available in the following databases:

- Chip-Seq data: Gene Expression Omnibus GSE46748 (<https://www.ncbi.nlm.nih.gov/geo/query/acc.cgi?acc=GSE46748>)

- Modeling computer scripts: GitHub (<https://github.com/SysBioChalmers/GECKO/releases/tag/v1.0>)

- [data type]: [full name of the resource] [accession number/identifier] [(doi or URL or identifiers.org/DATABASE:ACCESSION)]

- For data quantification: please specify the name of the statistical test used to generate error bars and P values, the number (n) of independent experiments (specify technical or biological replicates) underlying each data point and the test used to calculate p-values in each figure legend. The figure legends should contain a basic description of n, P and the test applied. Graphs must include a description of the bars and the error bars (s.d., s.e.m.).

- The References should be formatted according to the Molecular Systems Biology reference style (i.e. ordered alphabetically and listing the first 10 authors followed by et al).

- The information on the VALIGA investigators can be provided in the Appendix.

- Please include a Disclosure and Competing Interests statement in the main text.

- When you resubmit your manuscript, please download our CHECKLIST (<https://bit.ly/EMBOPressAuthorChecklist>) and include the completed form in your submission.

Please note that the Author Checklist will be published alongside the paper as part of the transparent process (<https://www.embopress.org/page/journal/17444292/authorguide#transparentprocess>).

Please resubmit your revised manuscript online, with a covering letter listing amendments and responses to each point raised by the referees. Please resubmit the paper ****within one month**** and ideally as soon as possible. If we do not receive the revised

manuscript within this time period, the file might be closed and any subsequent resubmission would be treated as a new manuscript. Please use the Manuscript Number (above) in all correspondence.

Click on the link below to submit your revised paper.

Link Not Available

Kind regards,

Maria

Maria Polychronidou, PhD
Senior Editor
Molecular Systems Biology

If you do choose to resubmit, please click on the link below to submit the revision online before 26th Oct 2023.

Link Not Available

IMPORTANT: When you send your revision, we will require the following items:

1. the manuscript text in LaTeX, RTF or MS Word format
2. a letter with a detailed description of the changes made in response to the referees. Please specify clearly the exact places in the text (pages and paragraphs) where each change has been made in response to each specific comment given
3. three to four 'bullet points' highlighting the main findings of your study
4. a 'standfirst text' summarizing in two sentences the study (approx. 250 characters)
6. a "thumbnail image" (width=211 x height=157 pixels, jpeg format), which can be used as 'visual title' to highlight your paper on our homepage.
7. Please include an author contributions statement after the Acknowledgements section (see <https://www.nature.com/msb/authors/index.html#Submission>)
8. When assembling figures, please refer to our figure preparation guideline in order to ensure proper formatting and readability in print as well as on screen:
<https://bit.ly/EMBOPressFigurePreparationGuideline>
See also figure legend guidelines: <https://www.embopress.org/page/journal/17444292/authorguide#figureformat>

*** PLEASE NOTE *** As part of the EMBO Publications transparent editorial process initiative (see our Editorial at <https://www.nature.com/msb/journal/v6/n1/full/msb201072.html>), Molecular Systems Biology will publish online a Review Process File to accompany accepted manuscripts. When preparing your letter of response, please be aware that in the event of acceptance, your cover letter/point-by-point document will be included as part of this File, which will be available to the scientific community. More information about this initiative is available in our Instructions to Authors. If you have any questions about this initiative, please contact the editorial office (msb@embo.org).

Reviewer #1:

The paper has improved considerably in terms of clarity and rigor. However, the presentation of the PILOT method should still be improved and the issues below resolved.

The introduction should end with a paragraph that describes the flow and main results of the paper.

Friedman-Nemenyi test: Was Nemenyi test performed after the Friedman test? I could not find the Friedman-Nemenyi in the github.

Fig. 1: Explain what p,q stand for. Also (as in my previous review) the notation j as both index and transport plan is problematic.

Fig. 2: "We observe that diffusion maps estimated with PILOT recovered trajectory-like structures in all analyzed data sets (Fig.2D-E;Sup. Fig. S2 and Sup. Fig.S3). Also, disease progression score is related with the severity of diseases in the data

sets (Fig. 2D-E; Sup. Fig. S4)". However methods show a similar behavior (e.g., Fig. 2E, S3, S4). PILOT does not seem to differentiate well between reduced and low disease states.

Fig. 3C: I guess each "bar" is actually circles that are too big and thus overlap. This should be fixed.

Does each circle represent a cell or cell type?

What do the colors mean, and what is the grey bar/circles?

The criteria for including the respective circles (i.e., cells/cell types) in the analysis and their coloring are unclear.

The expression of MYBPC3 does not seem to change with disease progression. Explain why it is claimed otherwise.

The kidney trajectory analysis (Fig. 4):

A pie chart describing the number of samples of control, reduced (intermediate state), or low filtration (disease state) should be added.

Fig. 4A shows that the most progressed samples according to PILOT are not the low, as should be, but rather the reduced ones. This does not seem to support the value of PILOT. Fig. 4C - same comment as Fig. 3C above.

Analysis of subgroups of pancreatic adenocarcinoma patients (Fig. 5):

Describe the number of distinct sample per group.

By what value was clustering was performed in Fig. 5A?

"Differential expression and GO analysis contrasting ductal cell 2 expression for Tumor 2 vs. Tumor 1 samples indicate the up-regulation of hypoxia related genes; and down-regulation of pancreas secretion genes (Fig. 5D-E)." I could not find these processes in Fig. 5E.

Were p-values in Fig. 5E,G adjusted for multiple hypothesis testing?

Discussion: "all most previous work" - all or most?

Minor:

The text contains several typos. Below are some examples:

"Area Under the Recall Precision Curve" - I assume this refers to area under the precision-recall curve

In page 4 AUPR is used instead of AUCPR in page 3.

Page 7 "course clustering" I guess you mean coarse clustering

Fig. 5A Y-axis titles should be fixed.

Reviewer #2:

Joodaki et al. addressed my previously raised comments. I have the following remaining comments.

1. Regarding batch effect, the authors have noticed that some data have strong batch effects and excluded data and samples (e.g. those generated by technologies other than 10X genomics) to analyze those from a particular tissue (e.g. lung) and from a uniform technique. Even with this, we are still assuming that data generated from a uniform technique and sampling procedure will not contain strong batch effects, and restricting the analysis to those samples will yield meaningful results. I believe a more proactive approach should be implemented in the package/framework to address the batch effect issue so as to avoid potentially misleading interpretations of the analysis output from PILOT.

2. Regarding my point 2 in the original comment, the authors provided additional heatmap visualizations of the OT distance matrices and their clustering in Supplementary Figure 1. However, I cannot find any reference and interpretation of these results in the revised manuscript. For some of the datasets/diseases, there is quite a high level of mixing (discordance) between class labels (i.e. disease/condition) compared to the clustering. Are some of these caused by technical variants/noise (e.g. batch effects) or driven by underlying biological signals?

3. The authors clarified that PILOT relies on the clustering of cells as input (not cell annotation from the original publication). Am I correct to assume that the original cell annotation is still used for computing the cost matrix C? The authors also mentioned that "...it defines a cost matrix (C) so that transporting masses between similar cell types (healthy vs. injured cardiomyocytes) have a lower cost than transporting masses between distinct cell types (cardiomyocytes vs. fibroblasts)". I cannot find specific details as to how the cost among similar and different cell types is determined.

Rev_Com_number: RC-2023-02028

New_manu_number: MSB-2023-11962

Corr_author: Costa

Title: Detection of Patient-Level distances from single cell genomics and pathomics data with Optimal Transport (PILOT)

Dear Poonam,

We would like to thank you and the reviewers for the careful consideration of the manuscript "Detection of Patient-Level distances from single cell genomics and pathomics data with Optimal Transport (PILOT)".

We have carefully reviewed the manuscript by considering reviewers and editorial requests. This review includes

- a new evaluation to support the fact PILOT trajectories do recover order of disease severity.
- Addition of functionality to PILOT to evaluate the presence of potential technical/confounding artifacts in the single-cell atlas data.

We have also done some changes in the manuscript wording and style as requested by you and reviewers.

We believe that the manuscript has further improved in quality and content during this review process and we are looking forward to your response.

For all authors, with best regards,

Ivan G. Costa

Reviewer #1:

The paper has improved considerably in terms of clarity and rigor. However, the presentation of the PILOT method should still be improved and the issues below resolved.

We sincerely appreciate the referee's insightful comments, which have enhanced the quality of our manuscript.

The introduction should end with a paragraph that describes the flow and main results of the paper.

We have added the text below in page 2 as suggested by the referee.

"In this study, we introduce PILOT (Patient level distance with Optimal Transport), which explores optimal transport for sample-based analysis of multi-scale single cell or pathomics data. First, we introduce PILOT's framework and its main methodological features. Next, we perform a benchmarking study to compare PILOT and competing approaches on their performance in 12 public single-cell and pathomics data sets. This indicated favorable results of PILOT in both clustering and trajectory prediction problems. Finally, we showcase PILOT's features by interpreting trajectory predictions on a single cell data set of samples with

myocardial infarction and a pathomics data set on samples with kidney disease; and clustering analysis of a single cell data with pancreatic adenocarcinoma patients.”

Friedman-Nemenyi test: Was Nemenyi test performed after the Friedman test? I could not find the Friedman-Nemenyi in the github.

Indeed, these are two tests, which are performed one after the other. The Friedman is a global test indicating if there is any significant difference between rankings of the algorithms and the Nemenyi is a post-hoc test to indicate significance in the difference of pairs of algorithms. The Nemenyi is only applied in case the null hypothesis of the Friedman test (no difference in ranks) is rejected. The code associated with the Friedman-Nemenyi test is found in <https://github.com/CostaLab/PILOT/blob/main/Tutorial/Friedman-Nemenyi.ipynb>. It is based on the tsutils package, which performs the Friedman test and Nemenyi tests seamlessly. We have improved the description of the test in the manuscript (see page 11).

Fig. 1: Explain what p,q stand for. Also (as in my previous review) the notation j as both index and transport plan is problematic.

We apologize for these recurring issues. We included definitions of p, l and q. We kept j as an index and adopted T as the transport plan. See page 2, 3, 6 and 7 and Fig. 1 for changes.

Fig. 2: "We observe that diffusion maps estimated with PILOT recovered trajectory-like structures in all analyzed data sets (Fig.2D-E;Sup. Fig. S2 and Sup. Fig.S3). Also, disease progression score is related with the severity of diseases in the data sets (Fig. 2D-E; Sup. Fig. S4)". However methods show a similar behavior (e.g., Fig. 2E, S3, S4). PILOT does not seem to differentiate well between reduced and low disease states.

We now include additional analysis and visualizations to address this issue.

First, for data sets with more than 2 classes (COVID-19 PBMC, Lung, Diabetes, Kidney/scRNA and Kidney IgAN), we now create an ordered variable, whose value increases with disease severity (control=1, mild=2, severe=3). Note that for the lung, we have classified chronic obstructive pulmonary disease (COPD) as mild and all carcinomas (SCLC, LA, NSCLC) as severe. For diabetes, we classified endocrine pancreas disorder as mild and type I and II diabetes as severe. The reasoning behind this is the difficulty in discriminating the true severity between types of diabetes or lung carcinomas based on the publicly available data, which limited clinical data describing these patients.

First, we include cumulative plots showing the proportion of control (blue), mild (orange) and severe (red) samples over pseudo-time. It can be observed that normal/control samples accumulate before mild and that mild cases accumulate before severe cases in all scenarios, with the exception of the diabetes data set where normal and mild are close together (**Reply Figure 1**).

Reply Fig. 1 - Cumulative probability of control, mild and severe cases over PILOT estimated disease progression (pseudotime).

To measure this more systematically, we estimated the Spearman Correlation (SC) between the estimated disease progression scores and the ordered values (control=1, mild=2, severe=3). For data sets with two classes, we adopt (control=1 and disease = 2). To verify ordering impacts predictions in 3 class data sets, we evaluated all possible orderings and we observed that using Control < Mild < Severe yields the highest SC values in all evaluated data sets (see **reply fig. 2**).

Reply Fig. 2: Spearman Correlation (y-axis) between disease progression scores and ordered classes for distinct data sets (x-axis) by using PILOT.

Altogether, these results support that PILOT disease progression orders samples in control, mild and severe cases. We include the Spearman Correlation in our benchmarking analysis (see new panel 2D; Appendix Table 2). We also included the re-ordering experiments and cumulative plots (Fig. EV2) and adapted the text of the manuscript (page 3-4).

Fig. 3C: I guess each "bar" is actually circles that are too big and thus overlap. This should be fixed. Does each circle represent a cell or cell type? What do the colors mean, and what is the grey bar/circles? The criteria for including the respective circles (i.e., cells/cell types) in the analysis and their coloring are unclear.

We have improved and simplified figure 3C following the referee's suggestions. Every dot represents a single cell, which is part of the target cluster. The color just reflects the expression level (y-axis). Despite being redundant, this helps the plot interpretation due to the high density of cells in some cases. The gray cells represented cells from other clusters (background). We decided to remove these for simplifying the plots and we only show the model fitted in these background cells (gray line). We improved the description of Fig. 3 legend and we believe that these changes make the plots more clear.

The expression of MYBPC3 does not seem to change with disease progression. Explain why it is claimed otherwise.

Indeed, there is only a marginal increase in the log scale expression of MYPBC3 from initial samples (3.2) towards the end of the trajectory (3.6), the model captures an up-ward trend (Rsquared of 0.55; p-value= 0.000000e+00), which also differs from the overall decrease of the expression of these genes in other cells (Wald Statistics 296, p-value= 0.000000e+00). We changed the text to indicate that we only observed a slight increase in expression of this gene. See page 5.

The kidney trajectory analysis (Fig. 4): A pie chat describing the number of samples of control, reduced (intermediate state), or low filtration (disease state) should be added.

We added a pie chart and exact sample sizes as legend of Fig. 4A.

Fig. 4A shows that the most progressed samples according to PILOT are not the low, as should be, but rather the reduced ones. This does not seem to support the value of PILOT. Fig. 4C - same comment as Fig. 3C above.

Indeed, at the diffusion map space, the end and start of the trajectories are less dense and this highlights outlier points noted by the referee. As discussed above, an alternative (and non-parametric) visualization is provided by the cumulative plot (**reply fig. 3**; also provided in new Figure EV2.) We believe that this additional analysis clarifies that PILOT can order samples regarding disease severity.

Reply Fig. 3 - Cumulative probability distribution (y-axis) of kidney IgAN data vs. predicted sample level disease progression (pseudotime - x-axis).

Analysis of subgroups of pancreatic adenocarcinoma patients (Fig. 5): Describe the number of distinct sample per group.

By what value was clustering was performed in Fig. 5A?

The clustering resolution value of 0.3 obtained the highest Silhouette score and identified three clusters: the first includes 11 control samples, and the remaining two are associated with PDAC samples comprising 14 samples classified as Tumor 1 and 10 samples as Tumor 2 (Fig. 5A). These details are now included in manuscript text (see page 5).

"Differential expression and GO analysis contrasting ductal cell 2 expression for Tumor 2 vs. Tumor 1 samples indicate the up-regulation of hypoxia related genes; and down-regulation of pancreas secretion genes (Fig. 5D-E)." I could not find these processes in Fig. 5E.

Hypoxia is indicated by the enrichment of *HIF-1 pathway* in Tumor 2 samples in Ductal cells 2 (Fig. 5E). Indeed, there is no indication of pancreas secretion alteration in Ductal cells II. This is rather present in stellate cells, which are shown in fig. 5F-G. We have improved the text to clarify this (see page 5).

Were p-values in Fig. 5E,G adjusted for multiple hypothesis testing?

All statistical tests presented in the manuscript were corrected for multiple testing. In the case of GO analysis, we have adopted the SCS algorithm proposed by the tool used for GO enrichment analysis (g:profiler; doi:[10.1093/nar/gkm226](https://doi.org/10.1093/nar/gkm226)). We included the term "adjusted p-value" in the legend of Fig. 5 and included details in the manuscript (page 9) to reflect this.

Discussion: "all most previous work" - all or most?

We meant and corrected to "all previous work".

Minor:

"Area Under the Recall Procession Curve" - I assume this refers to area under the precision-recall curve.

In page 4 AUPR is used instead of AUCPR in page 3.

Page 7 "course clustering" I guess you mean coarse clustering.

Fig. 5A Y-axis titles should be fixed.

We have implemented all these changes.

Reviewer #2:

Joodaki et al. addressed my previously raised comments. I have the following remaining comments.

1. Regarding batch effect, the authors have noticed that some data have strong batch effects and excluded data and samples (e.g. those generated by technologies other than 10X genomics) to analyze those from a particular tissue (e.g. lung) and from a uniform technique. Even with this, we are still assuming that data generated from a uniform technique and sampling procedure will not contain strong batch effects, and restricting the analysis to those samples will yield meaningful results. I believe a more proactive approach should be implemented in the package/framework to address the batch effect issue so as to avoid potentially misleading interpretations of the analysis output from PILOT.

We fully agree with the referee regarding the importance of dealing with batch effects. Indeed, this is one of the reasons why we focused on previously published single-cell data sets, where batch effects should have been previously considered. Moreover, the benchmarking analysis is done similarly for all evaluated approaches and highlights the ones with better recovery of the biological truth, i.e. disease labels, regardless of the presence of batch effects.

To empower PILOT users to evaluate the presence of batch effects, we now include statistical tests to evaluate the association between detected clusters or trajectories with any experimental or clinical variable provided in the data sets. For clustering analysis, we use the Chi-Squared statistic to compare results with discrete variables and ANOVA to compare with numerical variables. For trajectory analysis, we use ANOVA test to compare disease progression scores with discrete variables and Spearman Test to compare progression scores with numerical variables. For example, if we consider the Kidney scRNA-seq data, we observe a high association with the tissue location: renal medulla, cortex of the kidney, renal papilla, or kidney as a whole. We focus therefore on samples based on the unbiased sampling of cells (kidney). After filtering for these samples, we observe that the highest association of the estimated trajectory is with the disease type (or true labels). We also observe an association with diabetes status and BMI. This turns out to be spurious as all disease samples are diabetic and BMI are

only provided for controls. Similar results are obtained on the clustering analysis. We now include tutorials in PILOT (https://pilot.readthedocs.io/en/latest/Kidney_trajectory.html and https://pilot.readthedocs.io/en/latest/Kidney_clusters.html) and explicitly recommend users to do such analysis and in the manuscript (page 6 and 9). These additions enhance PILOT's ability to detect and account for potential batch effects, ensuring a more robust analysis.

2. Regarding my point 2 in the original comment, the authors provided additional heatmap visualizations of the OT distance matrices and their clustering in Supplementary Figure 1. However, I cannot find any reference and interpretation of these results in the revised manuscript.

These 12 data sets cover diverse organs and diseases, which we have low acquaintance with and lack access to full clinical data. We believe that a detailed analysis of these 12 data sets, which were used in the manuscript for benchmarking purposes, is out of the scope of a methodologically focused manuscript. Nevertheless, the manuscript still makes a more profound analysis of three of these data sets: pancreas with clustering analysis and kidney pathomics and myocardial infarction for the case of trajectories with respective interpretations.

For some of the datasets/diseases, there is quite a high level of mixing (discordance) between class labels (i.e. disease/condition) compared to the clustering. Are some of these caused by technical variants/noise (e.g. batch effects) or driven by underlying biological signals?

The question if the clusters are related to technical artifacts or biological signals is of course an important one. To address this, we use the statistical tests mentioned in point 1, to check if clusters are associated with either batch-related variables or any other clinical variable provided in the data (see new Appendix Table 3). For all 12 data sets, we observed a significant association of disease status (true labels) with the results, which confirmed the PILOT's performance and the relevance of these data sets for a benchmarking analysis. For most data sets (10 out of 12), the status variable (true labels) obtained the highest significance. For some data sets such as PDAC, Lupus, Kidney IgAN, we found several biological variables, which were associated with clusters such as diabetes status, staging, age, or sex. These associations were however mild and were not related to disease sub-clusters.

For the lung data set, which is composed of a meta analysis of several public data sets, we found that the study of origin is a potential batch variable, as it obtained a higher association (p-value $3.06e-25$) with the clusters as the disease "status" (p-value $9.32e-13$). For the COVID data set, the city of sample collection (p-value of $2.3e-09$) also showed a higher association than disease status (p-values $6.02e-04$). Despite these associations, we do not find any cluster driven by either technical variable alone (Reply Fig. 4). We were surprised by these results, as these labels and samples are used for biological analysis in the manuscripts reporting the data. Indeed, in the lung cells atlas (Sikkema et al. Nature Medicine, 2023) it was shown in Fig. 4 that some technical variables explain more variance in the data as the disease status. This analysis

did not include the study of origin. Regarding the COVID single-cell data (Ren et al. Cell, 2021), we could now find any reference to the city of origin in their analysis.

Reply Fig. 4 - Clustering of COVID (left) and Lung scRNA-seq data with batch related features.

Indeed, low dimensional visualizations and clustering analysis of samples in a standard approach for quality check in bulk RNA-seq (see for example DESeq2 tutorials; <https://bioconductor.org/packages/release/bioc/vignettes/DESeq2/inst/doc/DESeq2.html>). Such analysis cannot be easily performed for multiscale single-cell data. As demonstrated by our batch analysis, PILOT sample-based representations allow us to detect and characterize these potential issues and represent another potential contribution to the field. We expand our discussions in the main manuscript with these results. We try to raise the awareness that batch artifacts might be present in atlas-like studies and that PILOT provides tools to detect these. These results are integrated on page 4 and 6 of the manuscript.

3. The authors clarified that PILOT relies on the clustering of cells as input (not cell annotation from the original publication). Am I correct to assume that the original cell annotation is still used for computing the cost matrix C? The authors also motioned that "...it defines a cost matrix (C) so that transporting masses between similar cell types (healthy vs. injured cardiomyocytes) have a lower cost than transporting masses between distinct cell types (cardiomyocytes vs. fibroblasts)". I cannot find specific details as to how the cost among similar and different cell types is determined.

PILOT uses cell clusters as inputs. To estimate the cost matrices, centroids of these clusters are estimated and their distance (Eq. 7) is used as a cost function. This was described in the methods subsection "Estimation of the cost matrix". We changed the formulas and text in this section to make the association of Eq. 7 and the cost function C more explicit. PILOT does not require cluster labels in any estimation step, but their annotation to particular cell types or cell stages by the expert and this is crucial for interpretation. We revised some text passages to

reflect this and now use the term “annotated cell clusters” instead of “cell annotations” and “clusters” instead of “cell types” to avoid further confusion.

The discussion on the distance between cardiomyocytes and fibroblast cells is based on the cost matrix obtained at the analysis of the myocardial infarction. This was part of our tutorial (first panel in for trajectory https://pilot.readthedocs.io/en/latest/Myocardial_infarction.html) and is now included as Fig. EV1 and referenced in the text (page 2). There, we hope to more clearly show that cardiomyocyte cell types are similar to each other (healthy_CM, intermediary_CM, damaged_CM), while very distinct from fibroblast cells. Note that the cost function also defines some order within cell clusters: healthy cardiomyocytes (healthy_CM) are more like intermediary (low damage, intermediary_CM) than damaged cardiomyocyte cells (damaged_CM).

14th Nov 2023

Manuscript Number: MSB-2023-11962R

Dear Prof Costa,

Thank you for the submission of your revised manuscript to Molecular Systems Biology. I am pleased to inform you that we will be able to accept your manuscript pending the following final amendments:

1) Please check the "Author Checklist" carefully and complete all relevant questions. As you have ethical information in the manuscript for human research participants, please ensure these questions are answered in the checklist. Otherwise for each request on whether the information is included in the manuscript, please ensure that a choice has been made from the dropdown menu (i.e. please choose 'Not Applicable' for all the boxes that are not applicable, please do not leave them as 'Select Response').

2) In the main manuscript file, please do the following:

- Please include keywords (to max. 5).

- "Methods" and "Materials" should be combined into a single section called "Materials and Methods".

- Please include the following sentence in the Data availability section before the bullet points:

"The datasets and computer code produced in this study are available in the following databases:"

- In addition, please move the information in the 3rd bullet point on human research participant ethics into the relevant section in the Methods

- The manuscript sections should be in the following order: Title page - Abstract & Keywords - Introduction - Results - Discussion - Materials & Methods - Data Availability - Acknowledgments - Disclosure Statement & Competing Interests - References - Figure Legends - Tables with legends - Expanded View Figure Legends.

- Author contributions: Please remove it from the manuscript and specify author contributions in our submission system. CRediT has replaced the traditional author contributions section because it offers a systematic machine-readable author contributions format that allows for more effective research assessment. You are encouraged to use the free text boxes beneath each contributing author's name to add specific details on the author's contribution. More information is available in our guide to authors:

<https://www.embopress.org/page/journal/17574684/authorguide#authorshipguidelines>

3) For the figures and figure legends, please take care of the following:

- For Figure 2, panel E) is labeled twice, please check whether it should be F) the second time.

- Please remove all Expanded View figures from main manuscript file, but please leave the EV figure legends in the main manuscript (directly after the main manuscript figure legends).

- Please indicate the statistical test used for data analysis in the legends of figures 3b-c; 4b-c; 5d-g; EV3a-b

- Please note that the box plots need to be defined in terms of minima, maxima, centre, bounds of box and whiskers, and percentile in the legend of figures 2a-d; 5b.

- Please note that information related to n is missing in the legend of figure 5b.

4) Appendix file: In the Appendix file, please ensure the word "Appendix" is included in all labels for Appendix Figures and Appendix Tables including in the Table of Contents.

5) Please ensure that all funding sources are entered into the manuscript submission system (i.e. please add (DFG-GE 2811/3), Bundesministerium für Bildung und Forschung (BMBF e:Med Consortia Fibromap), German Research Foundation (DFG, Project IDs 322900939, 454024652, 432698239 445703531), European Research Council (ERC Consolidator Grant No 101001791), and the Federal Ministry of Education and Research (BMBF, STOP-FSGS-01GM2202C)).

6) Synopsis:

- Synopsis image: Please provide a synopsis image that summarises the main findings of the manuscript on a glance. The synopsis image should be uploaded as a high-resolution jpeg file 550 pixels wide x (250-400) pixels high. Please do not include a legend.

- Synopsis text: Please provide a short standfirst (maximum of 300 characters, including space), limit the bullet points to max. 5 and upload it as a separate .doc file. Please write the bullet points to summarise the key NEW findings. They should be designed to be complementary to the abstract - i.e. not repeat the same text. We encourage inclusion of key acronyms and quantitative information (maximum of 30 words / bullet point). Please use the passive voice.

7) As part of the EMBO Publications transparent editorial process initiative (see our Editorial at <http://embomolmed.embopress.org/content/2/9/329>), EMBO Molecular Medicine will publish online a Review Process File (RPF) to accompany accepted manuscripts. This file will be published in conjunction with your paper and will include the anonymous referee reports, your point-by-point response and all pertinent correspondence relating to the manuscript. Let us know whether you agree with the publication of the RPF and as here, if you want to remove or not any figures from it prior to publication. Please note that the Authors checklist will be published at the end of the RPF.

8) Please provide a point-by-point letter INCLUDING my comments as well as the reviewer's reports and your detailed responses (as Word file).

Click on the link below to submit your revised paper.

I look forward to reading a new revised version of your manuscript as soon as possible.

Yours sincerely,

Poonam Bheda, PhD
Scientific Editor
Molecular Systems Biology

Reviewer #1:

The authors have adequately addressed my concerns.
Some minor issues remain:
Fig. 2F should be renamed (2E appears twice).
Fig. 3C description of trend lines missing.
CSC algorithm should be SCS.
References were ordered by first name of the first author.

Reviewer #2:

The authors have addressed my comments. I note that still "Figure 1S" is not referenced anywhere in the main texts. Please refer to this Figure at least once at an appropriate place in the texts.

1) Please check the "Author Checklist" carefully and complete all relevant questions. As you have ethical information in the manuscript for human research participants, please ensure these questions are answered in the checklist. Otherwise for each request on whether the information is included in the manuscript, please ensure that a choice has been made from the dropdown menu (i.e. please choose 'Not Applicable' for all the boxes that are not applicable, please do not leave them as 'Select Response').

We have included the information in the author checklist. See lines 92-93 of "EMBO+Press+Author+Checklist".

2) In the main manuscript file, please do the following:

- Please include keywords (to max. 5).

This has been done (please see page 1).

- "Methods" and "Materials" should be combined into a single section called "Materials and Methods".

This has been done (please see page 6).

- Please include the following sentence in the Data availability section before the bullet points:

"The datasets and computer code produced in this study are available in the following databases:"

This has been done (please see page 11).

- In addition, please move the information in the 3rd bullet point on human research participant ethics into the relevant section in the Methods -

This has been done (please see page 10).

- The manuscript sections should be in the following order: Title page - Abstract & Keywords - Introduction - Results - Discussion - Materials & Methods - Data Availability - Acknowledgments - Disclosure Statement & Competing Interests - References - Figure Legends - Tables with legends - Expanded View Figure Legends.

This has been done.

- Author contributions: Please remove it from the manuscript and specify author contributions in our submission system. CRediT has replaced the traditional author contributions section because it offers a systematic machine-readable author

contributions format that allows for more effective research assessment. You are encouraged to use the free text boxes beneath each contributing author's name to add specific details on the author's contribution. More information is available in our guide to authors:

This has been done online.

3) For the figures and figure legends, please take care of the following:

- For Figure 2, panel E) is labeled twice, please check whether it should it be F) the second time.

This has been done (please see Figure 2).

- Please remove all Expanded View figures from main manuscript file, but please leave the EV figure legends in the main manuscript (directly after the main manuscript figure legends).

This has been done (please see page 16).

- Please indicate the statistical test used for data analysis in the legends of figures 3b-c; 4b-c; 5d-g; EV3a-b-

We have clarified the statistical tests related to each of these panels. See figure legends at pages 15-16.

- Please note that the box plots need to be defined in terms of minima, maxima, centre, bounds of box and whiskers, and percentile in the legend of figures 2a-d; 5b.

This has been done (please see legends of Figures 2 and 5 on page 15).

- Please note that information related to n is missing in the legend of figure 5b.

This has been done (please see the legend of Figure 5 on page 15).

4) Appendix file: In the Appendix file, please ensure the word "Appendix" is included in all labels for Appendix Figures and Appendix Tables, including in the Table of Contents.

This has been done (please see the Appendix).

5) Please ensure that all funding sources are entered into the manuscript submission system

This has been done online.

6) Synopsis:

- **Synopsis image:** Please provide a synopsis image that summarises the main findings of the manuscript on a glance. The synopsis image should be uploaded as a high-resolution jpeg file 550 pixels wide x (250-400) pixels high. Please do not include a legend.
- **Synopsis text:** Please provide a short standfirst (maximum of 300 characters, including space), limit the bullet points to max. 5 and upload it as a separate .doc file. Please write the bullet points to summarise the key NEW findings. They should be designed to be complementary to the abstract - i.e. not repeat the same text. We encourage inclusion of key acronyms and quantitative information (maximum of 30 words / bullet point). Please use the passive voice.
- **Please check your synopsis text and image before submission with your revised manuscript. Please be aware that in the proof stage minor corrections only are allowed (e.g., typos).**

The synopsis text (synopsis.docx) is part of the resubmission. We generated a few versions of the synopsis graph either using vector based pdf or a jpeg. Please use the one that fits your specifications.

7) As part of the EMBO Publications transparent editorial process initiative (see our Editorial at <http://embomolmed.embopress.org/content/2/9/329>), EMBO Molecular Medicine will publish online a Review Process File (RPF) to accompany accepted manuscripts. This file will be published in conjunction with your paper and will include the anonymous referee reports, your point-by-point response and all pertinent correspondence relating to the manuscript. Let us know whether you agree with the publication of the RPF and as here, if you want to remove or not any figures from it prior to publication. Please note that the Authors checklist will be published at the end of the RPF.

We agree with sharing the reply letters.

Reviewer #1:

The authors have adequately addressed my concerns.

Some minor issues remain:

Fig. 2F should be renamed (2E appears twice).

Fig. 3C description of trend lines missing.

CSC algorithm should be SCS.

References were ordered by first name of the first author.

Thank you for your detailed input. All the changes have been implemented in the manuscript.

Reviewer #2:

The authors have addressed my comments. I note that still "Figure 1S" is not referenced anywhere in the main texts. Please refer to this Figure at least once at an appropriate place in the texts.

This has been done (please see page 5).

24th Nov 2023

Manuscript number: MSB-2023-11962RR

Title: Detection of Patient-Level distances from single cell genomics and pathomics with Optimal Transport

Dear Prof Costa,

Thank you again for sending us your revised manuscript. We are now satisfied with the modifications made and I am pleased to inform you that your paper has been accepted for publication.

Yours sincerely,

Poonam Bheda, PhD
Scientific Editor
Molecular Systems Biology
